**Drivers of Phytoplankton Bloom Interannual Variability in the Amundsen and Pine**
**Island Polynyas**
**Guillaume Liniger[1,2*], Delphine Lannuzel[1,3,4], Sébastien Moreau[5,6], Michael S. Dinniman[7],**
**Peter G. Strutton[1,3]**
[1] Institute for Marine and Antarctic Studies, University of Tasmania, Hobart, Australia
[2] Monterey Bay Aquarium Research Institute, Moss Landing, CA, USA
[3] Australian Centre for Excellence in Antarctic Science, University of Tasmania, Hobart,
Australia
[4] Australian Antarctic Program Partnership, University of Tasmania, Hobart, Australia
[5] Norwegian Polar Institute, Tromsø, Norway
[6] iC3: Centre for ice, Cryosphere, Carbon and Climate, Department of Geosciences, UiT The
Arctic University of Norway, 9037 Tromsø, Norway
[7] Center for Coastal Physical Oceanography, Old Dominion University, Norfolk, VA, USA
* Corresponding Author: Guillaume Liniger (liniger@mbari.org)
**Abstract**
The Amundsen Sea Embayment (ASE) experiences both the highest ice shelf melt rates and the
highest biological productivity in West Antarctica. Using 19 years of satellite data and modelling
output, we investigate the long-term influence of environmental factors on the phytoplankton
bloom in the Amundsen Sea (ASP) and Pine Island (PIP) polynyas. We test the prevailing
hypothesis that changes in ice shelf melt rate could drive interannual variability in the polynyas'
surface chlorophyll-$a$ (chl$a$) and Net Primary Productivity (NPP). We find that the interannual
variability and long-term change in glacial meltwater may play an important role in chl$a$ variance
in the ASP, but not for NPP. Glacial meltwater does not explain the variability in neither chl$a$ or
NPP in the PIP, where light and temperature are the main drivers. We attribute this to potentially
greater amount of iron-enriched meltwater brought to the surface by the meltwater pump
downstream of the PIP, and the coastal ocean circulation accumulating and transporting iron
towards the ASP.
**Short Summary**
We investigate the phytoplankton bloom variability and its drivers in the Amundsen polynyas
(areas of open water within sea ice). Between 1998 and 2017, we find that changes in melting ice
shelves may have different impacts on biological productivity between the Amundsen Sea (ASP)
and Pine Island (PIP) polynyas. While ice shelves melting seems to play an important role for
phytoplankton growth variability in the ASP, light and warmer waters appear to be more
important in the PIP.
**1. Introduction**
Coastal polynyas are open ocean areas formed by strong katabatic winds pushing sea ice offshore
(Morales Maqueda, 2004). They are the most biologically productive areas in the Southern
Ocean (SO) relative to their size (Arrigo et al., 1998). This high biological productivity contrasts
sharply with the rest of the SO, where low iron and light availability generally co-limit
phytoplankton growth (Boyd et al., 2007). In West Antarctica, the Amundsen Sea Embayment
(ASE) hosts two of the most productive Antarctic polynyas: The Pine Island Polynya (PIP) and
Amundsen Sea Polynya (ASP) (Arrigo and van Dijken, 2003).

The phytoplankton community in the ASE is generally dominated by *Phaeocystis antarctica*
(Lee et al., 2017; Yager et al., 2016), which is adapted to low iron availability and variable light
conditions, and forms large summer blooms (Alderkamp et al., 2012; Yager et al., 2016).
Diatoms like *Fragilariopsis sp*. and *Chaetoceros sp.* are also present, often becoming more
important near the sea-ice edge or under shallow, stratified mixed layers where silicic acid (Si)
and iron (Fe) are more available (Mills et al., 2012). In exceptional years, such as 2020, diatoms
like *Dactyliosolen tenuijunctus* replaced *P. antarctica* as the dominant taxon, driven by
anomalously shallow mixed layers and sufficient Fe–Si supply (Lee et al., 2022). This dynamic
balance highlights how light, nutrient supply, and stratification control community composition
in these highly productive and complex Antarctic systems.

The ASE is also the Antarctic region experiencing the highest mass loss from the Antarctic ice
sheet. It has been undergoing increased calving, melting, thinning and retreat over the past three
decades (Paolo et al., 2015; Rignot et al., 2013; Rignot et al., 2019; Shepherd et al., 2018). In the
ASE, this ice loss is mainly through enhanced basal melting of the ice shelves. This is attributed
to an increase in wind-driven Circumpolar Deep Water (CDW) fluxes and ocean heat content
intruding onto the continental shelf through deep troughs such as the Pine Island and Dotson-
Getz, and flowing into the ice shelves cavities (Dotto et al., 2019; Jacobs et al., 2011; Pritchard et
al., 2012). There, warm waters fuel intense basal melt of the Pine Island, Thwaites, and Getz ice
shelves, and returns as a fresher, colder outflow that can strengthen stratification (Jenkins et al.,
2010; Ha et al., 2014). The PIP and ASP differ in their exposure to CDW and in local
circulation: the ASP is more strongly influenced by upwelled modified CDW (mCDW) and
glacial meltwater inputs, whereas in the PIP, the deep mCDW retains more of its original
offshore characteristics, with vertical exchange only significantly occurring beneath the ice
shelves, leading to a more stratified and less directly ventilated surface layer (Assmann et al.,
2013; Dutrieux et al., 2014). These hydrographic contrasts can shape the timing and magnitude
of phytoplankton blooms and nutrient dynamics across the two polynyas.

Melting ice shelves can explain about 60% of the biomass variance between all Antarctic
polynyas, suggesting that they are the primary supplier of dissolved iron (dFe) to coastal
polynyas (Arrigo et al., 2015), and can directly or indirectly contribute to regional marine
productivity (Bhatia et al., 2013; Gerringa et al., 2012; Hawkings et al., 2014; Herraiz-
Borreguero et al., 2016). The strong melting of the ice shelves can release significant quantities
of freshwater at depth (Biddle et al., 2017), resulting in a strong overturning within the ice
shelves cavity, called the meltwater pump (St-Laurent et al., 2017). Modelling efforts have
identified both resuspended Fe-enriched sediments and CDW entrained to the surface by the
meltwater pump as the two primary sources of dFe to coastal polynyas, providing up to 31% of
the total dFe, compared to 6% for direct ice shelves input (Dinniman et al., 2020; St-Laurent et
al., 2017). Other drivers such as sea-ice coverage (and associated increases in light and dFe
availability when sea ice retreats), or winds have also been shown to impact primary productivity
in polynyas (Park et al., 2019; Park et al., 2017; Vaillancourt et al., 2003).

The key question of how glacial meltwater variability may impact biological productivity in the
ASE has previously been raised during the ASPIRE program (Yager et al., 2012). During the
expedition, a significant supply of melt-laden iron-enriched seawater to the central euphotic zone
of the ASP was observed, potentially explaining why this area is the most biologically
productive in Antarctica (Randall-Goodwin et al., 2015; Sherrell et al., 2015). Other studies in
the Western Antarctic Peninsula and East Antarctica showed that the meltwater pump process
was also responsible for natural Fe supply to the surface, increasing primary productivity (Cape
et al., 2019; Tamura et al., 2023).

In this study, we investigate the long-term relationship between the main environmental factors
of the ASE and the surface biological productivity, with a focus on ice shelves melting. A
demonstrated relationship between glacial meltwater and phytoplankton growth would have far-
reaching consequences for regional productivity in coastal Antarctica, and possibly offshore,
over the coming decades under expected climate change scenarios (Meredith et al., 2019). We
test the hypothesis that changes in glacial meltwater are linked to the surface ocean primary
productivity variability observed over the last two decades. We use a combination of satellite
(ocean color and ice shelf melting rate), climate re-analysis, and model data spanning 1998 to

2017.


**2. Material and Methods**

2.1 Study area and polynya mapping

We focus on the PIP and ASP in the ASE in West Antarctica (Fig. 1). The ASE is comprised of
several ice shelves and glaciers, including: Abbot (Abb), Cosgrove (Cs), Pine Island (PIG),
Thwaites (Tw), Crosson (Cr), Dotson (Dt) and Getz (Gt). The PIG and Thwaites have received
significant attention in recent years due to their potentially large contribution to sea level rise
(Rignot et al., 2019; Scambos et al., 2017). Along with the Crosson and Dotson ice shelves, the
PIG and Thwaites are undergoing the highest melt rate, which is expected to increase under
climate change scenarios (Naughten et al., 2023; Paolo et al., 2023). The polynyas' boundaries
were determined using a 15% sea-ice concentration (SIC) mask (Moreau et al., 2015;
Stammerjohn et al., 2008) for every 8-day period from June 1998 to June 2017 to accurately
represent the size of the polynyas through time.

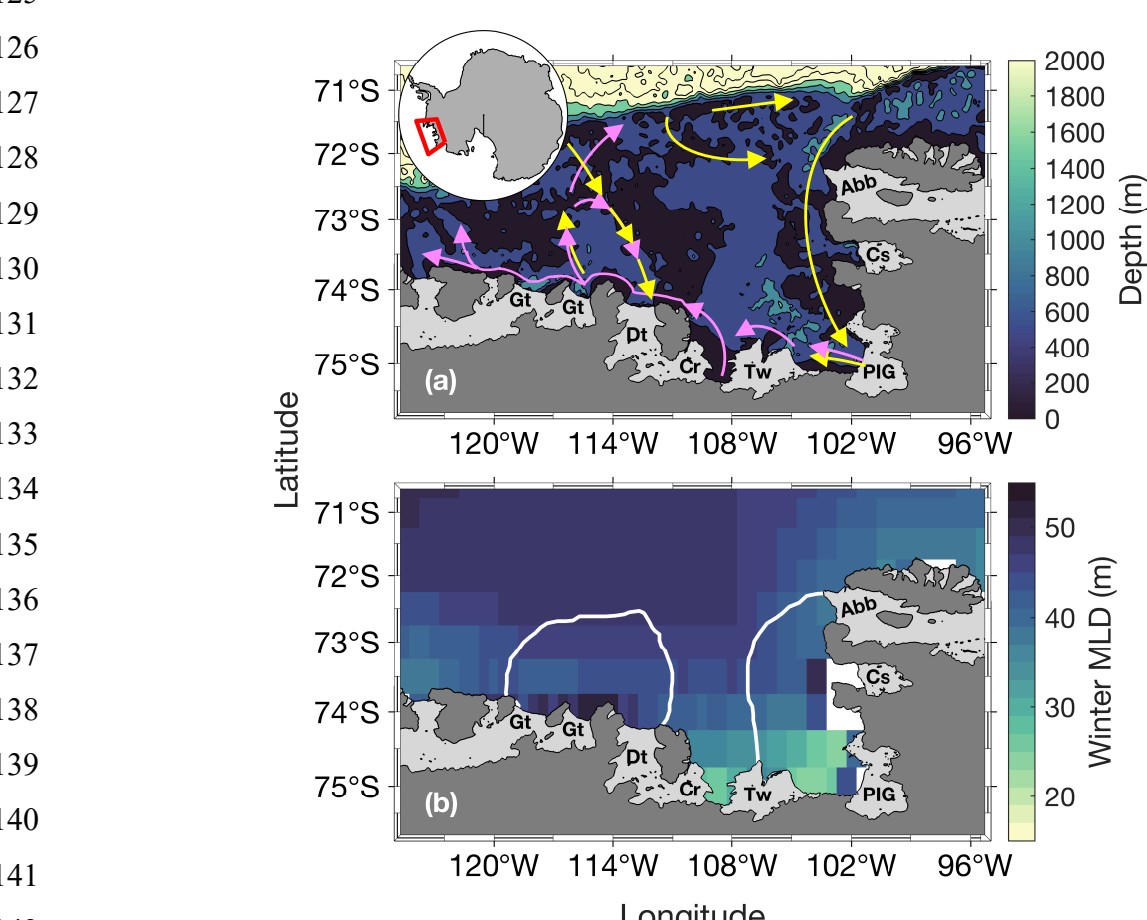

**Fig. 1.** Study area. Panel (a) shows the bathymetry (from ETOPO1; Amante & Eakins, 2009) and
panel (b) shows the climatological April-September (that we call winter) mixed-layer depth
(MLD) from 1998 to 2016 (n=114). Panel (a) shows a simplified schematic of the local deep
ocean circulation (~ below 400m, yellow arrows) and upper glacial
meltwater/sediments/circumpolar deep water sourced dFe pathways (magenta arrows), which
follows the local upper ocean circulation. Schematic adapted from St-Laurent et al. (2017). The
white lines in panel (b) represent the climatological summer polynyas' boundaries for the
Amundsen (ASP; left) and Pine Island (PIP; right) polynyas. The dark grey area is mainland
Antarctica. Light grey areas indicate floating ice shelves and glaciers: Abbot (Abb), Cosgrove
(Cs), Pine Island Glacier (PIG), Thwaites (Tw), Crosson (Cr), Dotson (Dt) and Getz (Gt).

2.2 Satellite ocean surface chlorophyll-*a* and net primary productivity


We obtained level-3 satellite surface chlorophyll-*a* concentration (chl*a*) with spatial and
temporal resolution of 0.04° and 8 days from the European Space Agency (ESA) Globcolor
project. We used the CHL1-GSM (Garver-Siegel-Maritorena) (Maritorena and Siegel, 2005)
standard Case 1 water merged products consisting of the Sea-viewing Wide Field-of-view
(SeaWiFS), Medium Resolution Imaging Spectrometer (MERIS), Moderate Resolution Imaging
Spectroradiometer (MODIS-A) and Visible Infrared Imaging Suite sensors (VIIRS). We chose to
perform our analysis with the merged GlobColour product, which has been widely applied and
tested in Southern Ocean and coastal Antarctic studies (Ardyna et al., 2017; Sari El Dine et al.,
2025; Golder & Antoine, 2025; Nunes, Fereira & Brito, 2025), to increase our spatial and
temporal coverage.

We estimated phytoplankton bloom phenology metrics following the Kauko et al. (2021)
method. Firstly, for a given 8-day period, we applied a spatial 3x3 pixels median filter to reduce
gaps in missing data. Then, if a pixel was still empty, we applied the average chl*a* of the previous
and following week to fill the data gap. Data were smoothed using a 4-point moving median
(representing a month of data). For each pixel, the threshold for the bloom detection was based
on 1.05 times the annual median. The threshold method is frequently used (Racault et al., 2012;
Siegel et al., 2002) and proven reliable at higher latitudes (Marchese et al., 2017; Soppa et al.,
2016; Thomalla et al., 2023). We then determined 5 main bloom metrics. The bloom start (BS) is
defined as the day where chl*a* first exceeds the threshold for at least 2 consecutive 8-day periods.
Conversely, the bloom end is the day where chl*a* first falls below the threshold for at least 2
consecutive 8-day periods. The bloom duration (BD) is the time elapsed between bloom start and
bloom end. The bloom mean chl*a* (BM) and bloom maximum chl*a* are respectively the average
and maximum chl*a* value calculated during the bloom. Each year is centered around austral
summer, from June 10[th] year *n* (day 1) to June 9[th] year *n+1* (day 365 or 366). We also averaged
our 8-day data to monthly data to perform a spatial correlation analysis (see section 2.6).

We note that satellite ocean-colour chl*a* algorithms (including the GlobColour merged product
used here) are globally tuned and may underperform in optically complex waters (e.g., with
elevated dissolved organic matter or suspended sediments, 'Case 2'). In the ASP, past work
(Park et al. 2017) showed that satellite chl*a* climatologies reflect broad seasonal patterns that are
consistent with *in situ* measurements of phytoplankton biomass and photophysiology, but there is
limited data from regions immediately adjacent to glacier fronts or during times of strong
meltwater input. Thus, while we consider satellite chl*a* to be useful for capturing spatial and
temporal variability at polynya scale, uncertainty likely increases in optically complex zones
near glacier margins or during low-light periods, and needs to be considered while interpreting
results.

Eight-day satellite derived Net Primary Productivity (NPP) data with 1/12° spatial resolution,
spanning 1998 - 2017 using the Vertically Generalized Production Model (Behrenfeld and
Falkowski, 1997) were obtained from the Oregon State University website. The VGPM model is
a chlorophyll-based approach and relies on the assumption that NPP is a function of chl*a*,
influenced by light availability and maximum daily net primary production within the euphotic
zone. SeaWiFS-based NPP data span 1998 - 2009, MODIS-based data span 2002 - 2017. To
increase spatial and temporal coverage, we averaged SeaWiFS and MODIS from 2002 to 2009,
where there was valid data for both in a pixel. NPP data were also monthly averaged and used to
compare with chl*a* spatial and temporal patterns.

We caution that our study focuses on surface productivity, and satellites cannot detect under-ice
phytoplankton, sea-ice algal blooms, or deeper productivity, therefore likely underestimating
total primary productivity (Ardyna et al., 2020; Boles et al., 2020; Douglas et al., 2024; McClish
& Bushinsly, 2023; Stoer & Fennel 2024).

2.3 Ice shelves volume flux


We used the latest ice shelf basal melt rate estimates from Paolo et al (2023). These estimates are
derived from satellite radar altimetry measurements of ice shelves height, and produced on a 3
km grid every 3 months, with an effective resolution of ~5 km. For this study, our basal melt
record spans June 1998 to June 2017. We calculated ice shelves volume flux rate for every
gridded cell by multiplying the basal melt rate by the cell area. Data were summed for each ice
shelf for a 3-month period. A 5-point (15 months) running mean was applied to reduce noise,
such as spurious effects induced by seasonality on radar measurements over icy surfaces (Paolo
et al., 2016), and data were temporally averaged from October to March to match the SO
phytoplankton growth season (Arrigo et al., 2015), providing yearly mean values. The Abbot,
Cosgrove, Thwaites, PIG, Crosson, Dotson and Getz ice shelves were used to calculate a single
total meltwater volume flux (TVFall) for the ASE to investigate the link with surface chl*a* and
NPP. We also investigated the relationship between each polynyas' productivity and their closest
ice shelf. The Abbot, Cosgrove, PIG and Thwaites ice shelves were used to calculate the flux
rate in the PIP (TVFpip) while the Thwaites, Crosson, Dotson and Getz ice shelves were chosen
for the ASP (TVFasp). The Thwaites was used in both due to its central position between the two
polynyas. We thereafter use the term glacial meltwater which defines meltwater resulting from
ice shelf melting.

2.4 Simulated dFe distribution


The spatial distribution of dFe from different sources in the embayment was investigated from
Dinniman et al. (2020) model output. The model used is a Regional Ocean Modelling System
(ROMS) model, with a 5 km horizontal resolution and 32 terrain following vertical layers and
includes sea-ice dynamics, as well as mechanical and thermodynamic interaction between ice
shelves and the ocean. The model time run spans seven years and simulates fourteen different
tracers to understand dFe supply across the entire Antarctic coastal zone, with the last two years
simulating biological uptake. For the purpose of this study, we only use four different dFe
sources/tracers in the ASE: ice shelf melt, CDW, sediments and sea ice. Each tracer estimation is
independent from each other, meaning that one source does not affect the other, and they have
the same probability for biological uptake by phytoplankton. That is, dFe from all sources can
equally be taken up by phytoplankton. This is parametrized in the model as all iron molecules
being bound to a ligand and therefore remaining in solution in a bioavailable form (Gledhill &
Buck, 2012). For a detailed and complete explanation of the model, see Dinniman et al. (2020).

2.5 Other environmental parameters


We used SIC data spanning June 1998 to June 2017 from the National Snow and Ice Data Center
(Cavalieri et al., 1996). The data are Nimbus-7 SMMR and SSMI/SSMIS passive microwave
daily SIC with 25 km spatial resolution. We computed the sea-ice retreat time (IRT) and open
water period (OWP) metrics using a 15% threshold (Stammerjohn et al., 2008). Daily data were
monthly averaged to perform a spatial correlation analysis (see section 2.6).

We collected monthly level-4 Optimum Interpolation Sea Surface Temperature (OISST.v2)
0.25° high resolution dataset from the National Oceanic and Atmospheric Administration
(Huang et al., 2021). Using this dataset compared to others has been proven to be the most
suitable for our region of interest (Yu et al., 2023).

We obtained monthly Photosynthetically Available Radiation (PAR) from the same Globcolour
project at the same spatial and temporal resolution (0.04° and 8 days) as chl*a*.

We used monthly averaged ERA5 reanalysis of zonal (u) and meridional (v) surface wind speed
at 10 m above the surface (Hersbach et al., 2020).

We investigated monthly mean MLD from the Estimating the Circulation and Climate of the
Ocean (ECCO) ocean and sea-ice state estimate project (ECCO consortium et al., 2021). The
dataset is the version 4, release 4, at 0.5° spatial resolution.

Variability in the sea-ice landscape can be influenced by the Amundsen Sea Low (ASL) in West
Antarctica (Hosking et al., 2013; Turner et al., 2016). We therefore finally looked at the impact
of the ASL and its potential influence on sea-ice variability. Monthly ASL indices (latitude,
longitude, central and sector pressure) derived from ERA5 reanalysis data were obtained from
the ASL climate index page (Hosking et al., 2016).


276   2.6 Statistical analysis


278 Because some of our data were not normaly distributed, we consistently applied nonparametric

279 tests throughout our statistical analysis. A Mann-Kendall test was performed to detect linear

280 trends in chl*a* and NPP. A two-tailed non-parametric Spearman correlation metric (rho, *p*) was

281 calculated to investigate the relationship between chl*a*, NPP, and environmental factors, as well

282 as between the phytoplankton bloom and sea-ice phenology metrics. A two-tailed Mann-Whitney

283 test was performed to detect any significant mean differences for chl*a,* sea-ice phenology

284 metrics, MLD, PAR and dFe sources between the two polynyas. Monthly spatial correlations

285 were tested between SIC, winds, chl*a*, NPP, SST, and PAR after removing the seasonality for

286 each parameter. As well, a yearly spatial correlation between chl*a*, NPP and TVFall was

287 performed. The relationships between chl*a*, NPP and environmental factors were explored using

288 a Principal Component Analysis (PCA). No pre-treatment (mean-centering or normalization)

289 was applied to the variables prior to PCA, as all variables are expressed in comparable units and

290 ranges, consistent with common practice in marine biogeochemistry studies (Marchese et al.,

291 2017; Liniger et al., 2020). The Spearman, Mann-Whitney and PCA analysis were conducted

292 using the mean TVFs, MLD, SST, and PAR calculated over the October-March period for each

293 year, with the associated bloom and sea-ice phenology metrics. Every statistical test was run with

294 a 95% (p-value < 0.05) confidence level. Our study spans 1998-2017. We are constrained by the

295 start of satellite ocean color data (1998) and the end of the ice shelf basal melt rate record (2017)

296 from Paolo et al (2023).

298 **3. Results**

300   3.1 Glacial meltwater, chl*a*, and NPP variability

302 The annual climatology maps reveal substantially higher chl*a* and NPP in the ASP compared to

303 the PIP (Fig. 2). Chl*a* starts increasing in mid-November to reach its average peak earlier in the

304 PIP than the ASP. At its peak, chl*a* in the ASP is 6.49 mg m$^{-3}$ and 4.94 mg m$^{-3}$ in the PIP (Fig.

305 3a). During the bloom period, chl*a* is also higher in the ASP on average compared to the PIP

306 (ASP = 5.21 ± 1.29 mg m$^{-3}$; PIP = 3.69 ± 1.11 mg m$^{-3}$; p-value < 0.01; Fig. 3b; Supplementary

Table T1). When looking at polynya area integrated values (concentration multiplied by area
gives units of mg m$^{-1}$), chl$a$ is significantly higher in the ASP than in the PIP, and increases with
the polynya area (Supplementary Figs. S1 and S2). NPP is also significantly higher in the ASP
than in the PIP (1.88 ± 1.12 TgC y$^{-1}$ vs 0.85 ± 0.86 TgC y$^{-1}$, p-value = 0.004; Supplementary
Fig. S3). No significant interannual trends in mean chl$a$ and NPP during the bloom are observed
for either polynya (p-value > 0.1; Fig. 3b; Supplementary Fig. S3). The climatological winter
MLD in the ASP is deeper (MLD ASP = 45.8 ± 8.0m; MLD PIP = 36.4 ± 7.3m; p-value < 0.01;
Fig. 1b), indicating that it may better entrain deeper sources of nutrients into the upper waters for
the following phytoplankton growing season, resulting in higher productivity (Fig. 2).

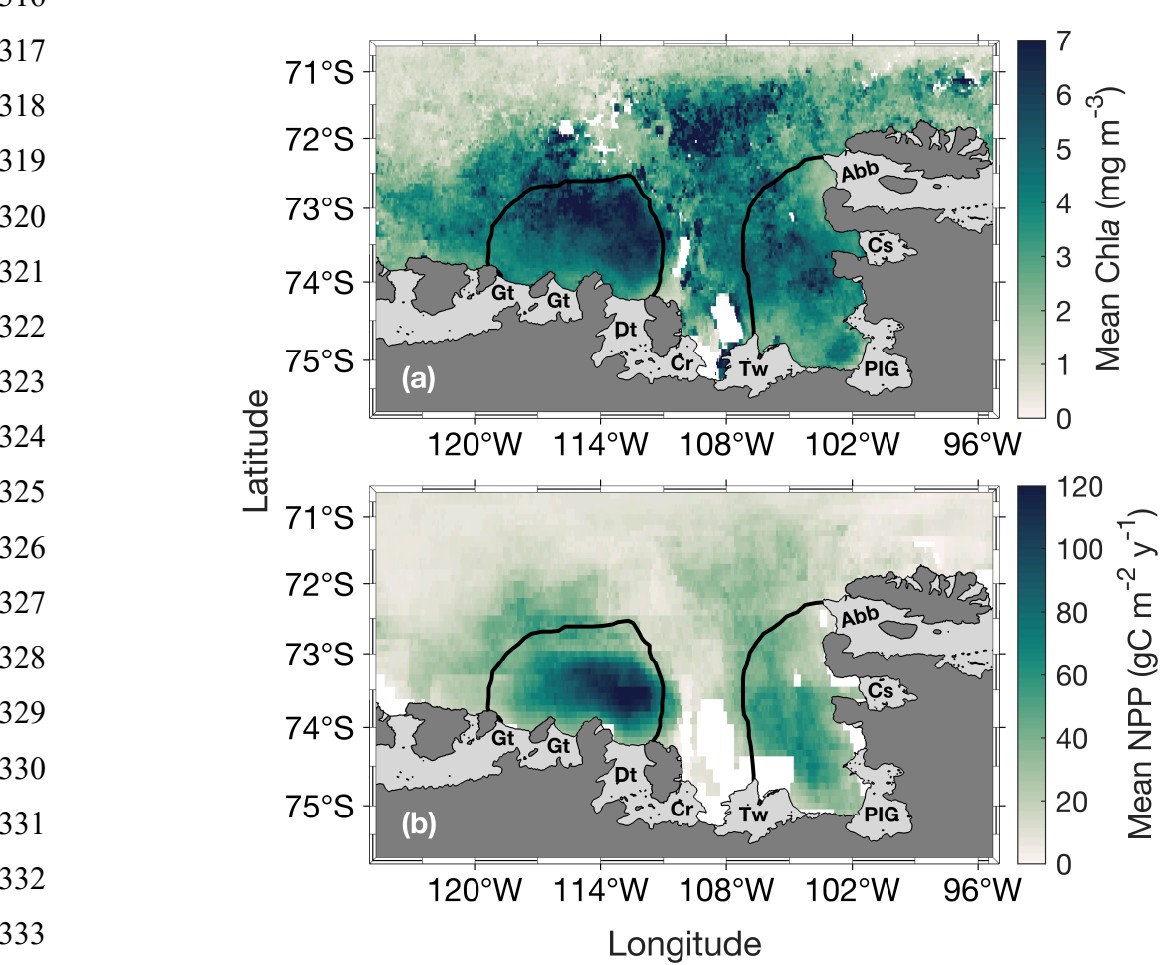

**Fig. 2.** Spatial distribution of (a) mean surface chlorophyll-$a$ (chl$a$) concentration during the
bloom and (b) net primary productivity (NPP) climatology (1998 – 2017) for the Amundsen
(ASP; left) and Pine Island (PIP; right) polynyas. The black lines represent the climatological
summer polynyas' boundaries.

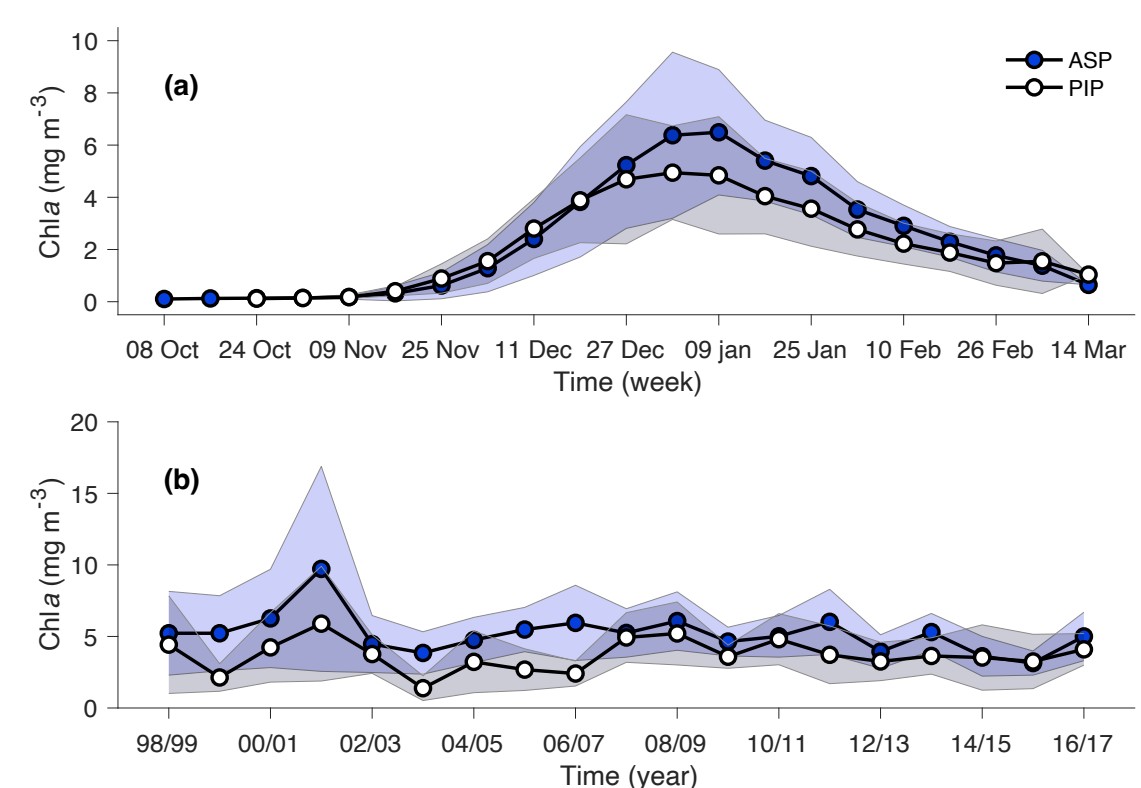

**Fig. 3.** (a) Weekly chlorophyll-*a* (chl*a*) concentration climatology (1998-2017) for the

Amundsen (ASP; blue circles) and Pine Island (PIP; white circles) polynyas. (b) Bloom mean

chl*a* time series of ASP (blue circles) and PIP (white circles). Shadded areas resrepsent the

standard devation for a given year. The relationship between chl*a* (in mg m$^{-3}$ and mg m$^{-1}$) and the

polynya size is shown in Supplementary Fig. S2.

The variability in TVFall is statistically uncorrelated with surface chl*a* and NPP in both polynyas

from 1998 to 2017 (Fig. 4; Supplementary Fig. S4). However, the relationship becomes strongly

significant in the ASP for both mean and maximum chl*a* when we remove the chl*a* outlier in

2001/02 (red data point; Figs. 4a-b), although not for NPP (Supplementary Figs. S4a-b). The

positive relationship implies that surface chl*a* in the ASP is higher when more glacial meltwater

is delivered to the embayment. No strong relationships are observed in the PIP between TVFall,

surface chl*a* and NPP (Figs. 4c-d; Supplementary Figs. S4c-d).

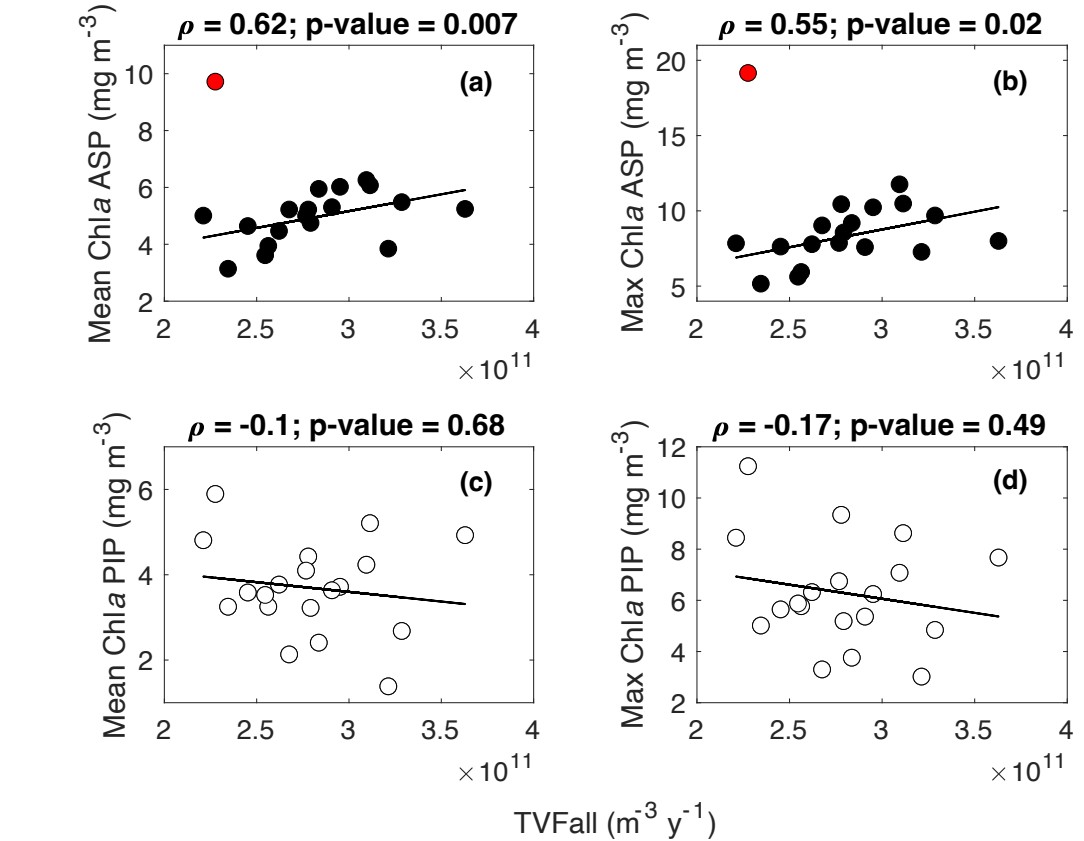

**Fig. 4.** Scatter plots of mean and maximum (max) surface chlorophyll-*a* (chl*a*) concentrations
with the total volume flux (TVFall) for (a-b) the Amundsen (ASP) and (c-d) the Pine Island
(PIP) polynyas from 1998 to 2017 (n=19). The fitted lines and statistics exclude year 2001/02
(red outlier) for the ASP regressions. If all data is considered, the relationships between mean
chl*a*, max chl*a* and TVFall in the ASP are not significant. TVFall is an annual integral
representing the sum of all ice shelves (see methods section) for the Amundsen Sea Embayment
(ASE).

When fluxes from individual glaciers are considered, PIP chl*a* does not correlate with Abbot,
Cosgrove, PIG, Thwaites or TVFpip fluxes (Table 1). On the other hand, ASP chl*a* shows strong
relationships with TVFasp, the Dotson and Crosson ice shelves (Table 1), and all ice shelves

become significantly correlated with mean and maximum chl*a* when year 2001/02 is removed.
There are no statistically significant relationships between individual ice shelves and NPP in
either polynyas.

**Table 1.** Statistical summary (Spearman's rank correlation) of the relationships between ice
shelves volume flux, mean and maximum (max) surface chlorophyll-*a* (chl*a*) concentrations
(n=19) in both polynyas. The * marks a significant (p-value < 0.05) relationship. All relationships
between mean chl*a*, maximum chl*a* and ASP ice shelves become significant when year 2001/02 is
removed.


| | Amundsen Sea polynya (ASP) | | | | Pine Island polynya (PIP) | | | |
|---|---|---|---|---|---|---|---|---|
| | Mean chl*a* | | Max chl*a* | | Mean chl*a* | | Max chl*a* | |
| | rho | p-value | rho | p-value | rho | p-value | rho | p-value |
| Abbot | / | / | / | / | 0.09 | 0.73 | -0.04 | 0.88 |
| Cosgrove | / | / | / | / | -0.32 | 0.18 | -0.46 | 0.05 |
| PIG | / | / | / | / | -0.04 | 0.88 | -0.13 | 0.61 |
| Thwaites | 0.16 | 0.52 | 0.11 | 0.66 | 0.12 | 0.63 | 0.09 | 0.71 |
| Crosson | 0.43 | 0.07 | **0.50** | **0.03*** | / | / | / | / |
| Dotson | **0.48** | **0.04*** | **0.54** | **0.02*** | / | / | / | / |
| Getz | 0.37 | 0.12 | 0.43 | 0.07 | / | / | / | / |
| TVFasp | 0.42 | 0.07 | **0.46** | **0.05*** | / | / | / | / |
| TVFpip | / | / | / | / | 0.009 | 0.97 | -0.1 | 0.68 |

Spatially, the mean and maximum chl*a* are strongly correlated with TVFall in southern-eastern part of the ASP, in front of the Dotson ice shelf (Figs. 5a-b), where a positive relationship with NPP is also observed (Fig. 5c), although not significant.

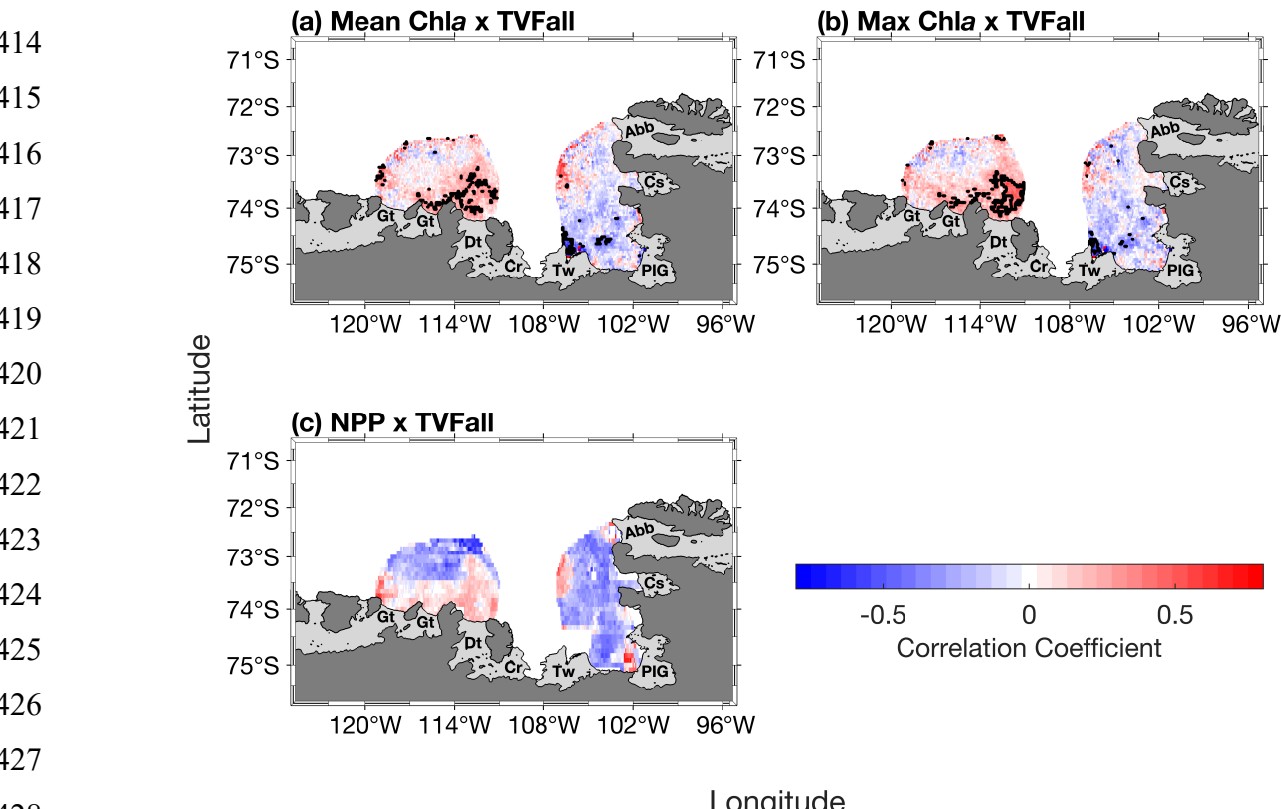

**Fig. 5**. Spatial correlation maps between total volume flux (TVFall) and (a) surface mean chlorophyll-*a* (chl*a*) concentration, (b) surface maximum (max) chl*a* concentration and (c) net primary productivity (NPP) (n=19). The black contour represents significant correlations at 95% confidence level. Data outside of the summer climatological polynyas' boundaries were masked out.

### 3.2 Simulated dFe sources distribution

The modelled spatial distribution of surface dFe sources is presented in Fig. 6. On average, the smallest dFe source in the embayment is from the ice shelves, with a maximum concentration

between the Thwaites and Dotson ice shelves. The dFe from sea ice is slightly higher than from ice shelves and similar over the two polynyas, and is higher near the sea-ice margin (Fig. 6b). The dFe from CDW is also higher between the Thwaites and Dotson (Fig. 6c). Sediment is the dominant dFe source (Fig. 6d). Its distribution spreads from 108°W to the western part of the Getz ice shelf. The highest sediment-sourced dFe concentration is found along the coast and inside the ASP. On polynya-wide average basis, the sediment reservoir contributes significantly more to total dFe in the ASP (58.3%, 0.13nM) compared to sea ice (16.5%, 0.04nM), CDW (13.5%, 0.03nM) and ice shelves (11.7%, 0.03nM). In the PIP, the contribution of sediments is still significantly higher (41.2%; 0.08nM) but lower than the ASP and the contribution gap with the other sources decreases. The CDW and sea ice contribute 22.5% (0.04nM) and 18.9% (0.035nM) to the dFe pool respectively, while ice shelves are still the smallest sources at 14.5% (0.03nM) in the PIP.

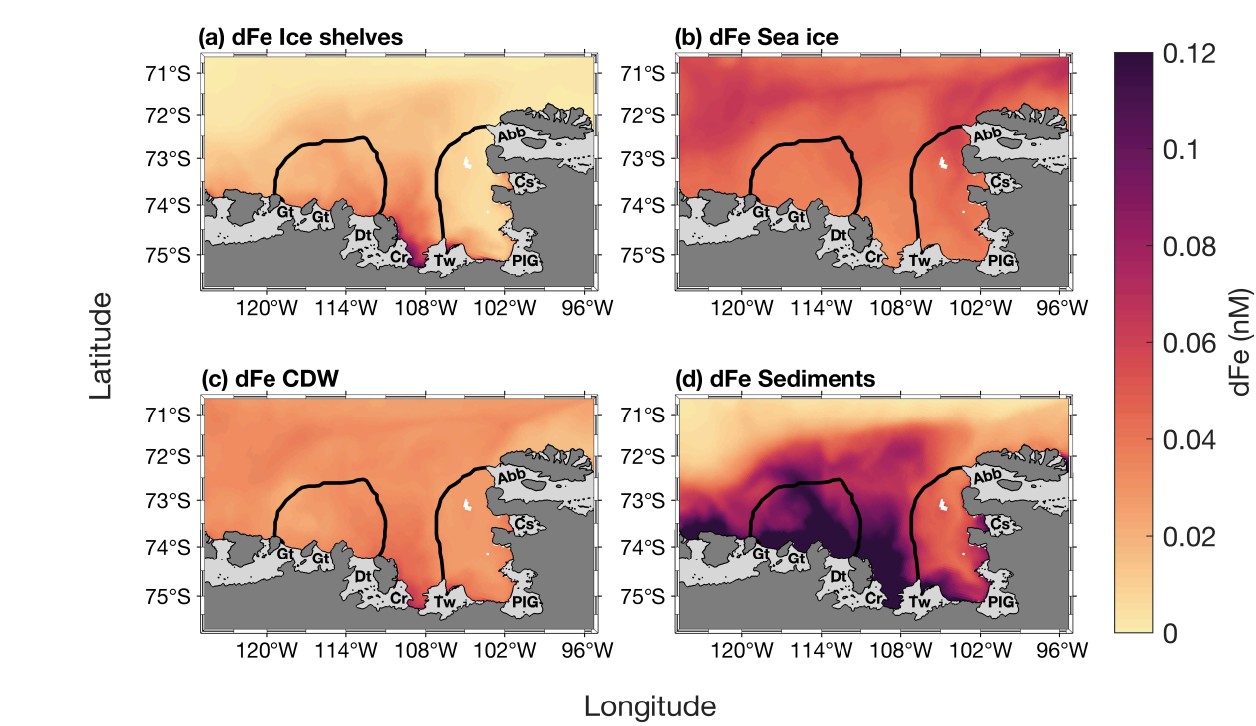

**Fig. 6.** Two-years top-100m averaged spatial distribution of surface dissolved iron (dFe) contribution from (a) ice shelves, (b) sea ice, (c) circumpolar deep water (CDW) and (d) sediments simulated by the model from Dinniman et al. (2020). The black lines represent the climatological summer polynyas' boundaries.

3.3 Environmental parameters, chl*a* and NPP variability


During the phytoplankton growth season (October-March), SIC is spatially significantly
anticorrelated to the meridional winds speed in both polynyas (Fig. 7a). Chl*a* is significantly
positively correlated with SST in the eastern ASP, and the whole PIP (Fig. 7b), but weakly with
PAR in both polynyas (Fig. 7c). Finally, PAR and SST are positively related in both central
polynyas, albeit not significantly (Fig. 7d). We note that similar spatial relationships are
observed when NPP is correlated with SST and PAR (Supplementary Fig. S5).















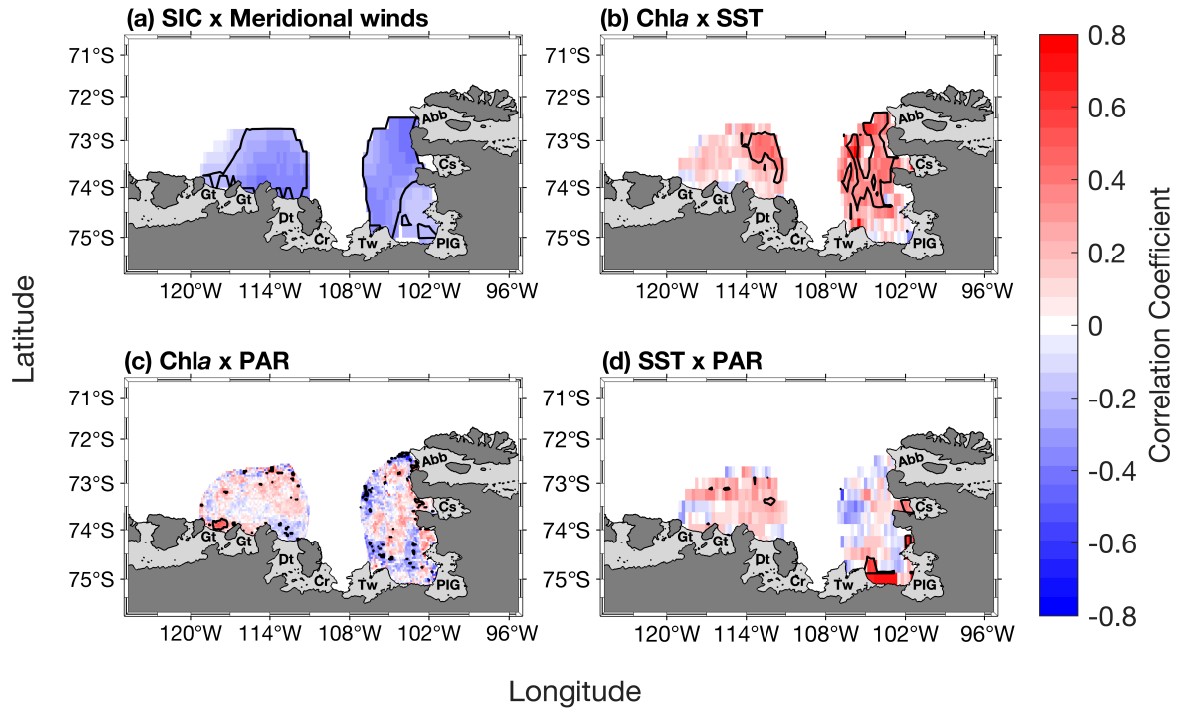

**Fig. 7**. Spatial correlation map between sea-ice concentration (SIC) and (a) meridional winds.
Spatial correlation maps between mean chlorophyll-*a* (chl*a*) concentration and (b) sea surface
temperature (SST), (c) photosynthetically available radiation (PAR). (d) Spatial correlation map
between PAR and SST. Data span 1998 – 2017 from October to March (n=114). The black contour
represents significant correlations at 95% confidence level. Seasonality was removed before
performing the correlation. Data outside of the summer climatological polynyas' boundaries were
masked out.

Regarding the phenology, the bloom start is positively correlated to IRT and negatively with
OWP in the ASP, although not significantly with the OWP (Table 2). This means that the bloom
starts earlier and later as IRT does, and that longer OWP and earlier bloom starts are correlated
with earlier ice retreat. The bloom mean and bloom maximum (max) chl*a* are not correlated with
either IRT and OWP in the ASP. IRT and OWP are significantly related ($p = -0.93$; p-value <
0.001). When year 2001/02 is removed, no significant changes in the relationships between all
parameters are detected. In the PIP, all metrics are significantly related to each other, except for
PAR and OWP (Table 2). That is, the bloom start is positively correlated with IRT and
negatively with OWP, while the bloom duration, mean chl*a*, max chl*a* and NPP are negatively
linked to the IRT and positively with OWP. SST and PAR are negatively correlated with IRT,
and positively with SST. IRT and OWP are significantly related in the PIP ($p = -0.88$; p-value <
0.001).


**Table 2.** Statistical summary (Spearman's rank correlation) of the relationships between the sea-
ice phenology metrics and environmental parameters (n=19) in both polynyas. The * marks a
significant (p-value < 0.05) relationship. IRT = ice retreat time, OWP = open water period, NPP =
net primary productivity, SST = sea surface temperature, PAR = photosynthetically available
radiation. Removing year 2001/02 for the ASP slightly changes the strength of the relationships
between parameters (i.e., rho) but not the significance.

| | Amundsen Sea polynya (ASP) | | | | Pine Island polynya (PIP) | | | |
|---|---|---|---|---|---|---|---|---|
| | IRT | | OWP | | IRT | | OWP | |
| | rho | p-value | rho | p-value | rho | p-value | rho | p-value |
| Bloom start | **0.51** | **0.03*** | -0.43 | 0.07 | **0.56** | **0.02*** | **-0.48** | **0.04*** |
| Bloom duration | -0.12 | 0.63 | 0.09 | 0.71 | **-0.56** | **0.02*** | **0.59** | **0.01*** |
| Bloom mean | 0.19 | 0.44 | -0.33 | 0.17 | **-0.67** | **0.003*** | **0.50** | **0.04*** |
| Bloom max | 0.24 | 0.32 | -0.35 | 0.14 | **-0.65** | **0.005*** | **0.52** | **0.03*** |
| NPP | **-0.55** | **0.02*** | 0.45 | 0.05 | **-0.72** | **0.001*** | **0.54** | **0.02*** |

| | | | | | | | | |
|---|---|---|---|---|---|---|---|---|
| SST | -0.09 | 0.72 | -0.01 | 0.96 | **-0.57** | **0.02*** | **0.52** | **0.03*** |
| PAR | -0.09 | 0.72 | 0.05 | 0.84 | **-0.62** | **0.007*** | 0.38 | 0.12 |

We explore the relationships between phytoplankton bloom phenology metrics and their potential environmental drivers by conducting a multivariate PCA for both polynyas (Fig. 8). The PCA reduces our datasets (11 variables) and breaks them down into dimensions that capture most of the variability and relationships between all variables. Arrows indicate the contribution of each variable to the dimensions, with longer arrows representing stronger influence. Observations (in our case, years) positioned in the direction of an arrow are more influenced by that variable. In the ASP (Fig. 8a), the first two principal components explain 55.3% of the total variance (Dim1: 35%, Dim2: 20.3%). NPP in the ASP is closely associated with BD, indicating that the bloom duration is the primary driver of production. On the other hand, environmental vectors such as TVFall and TVFasp project more strongly onto Dim2 with the bloom mean chl*a*, indicating that meltwater input may influence surface chl*a* interannual variability, and is less directly tied to NPP. We note that when year 2001/02 is removed, the relationship between TVFasp and TVFall becomes much stronger with BM (Supplementary Fig. S6a) and is slightly anticorrelated to SST and MLD. In the PIP (Fig. 8b), the first two components account for 63.7% of the total variance (Dim1: 46.7%, Dim2: 17%). Compared to the ASP, both NPP and BM cluster strongly with BD and PAR on Dim1. Additionally, IRT, OWP and SST and MLD aligned along Dim1, which explains 46.7% of the total variance compared to 35% for the ASP, suggesting that physical conditions might play a stronger structuring role in PIP compared to the ASP. In contrast, TVFall and TVFpip stand alone and align more strongly with Dim2, suggesting a less dominant influence of meltwater on BM and NPP variability in the PIP. The summer polynya-averaged PAR and MLD are significantly stronger and deeper, respectively, in the ASP compared to the PIP during the bloom season (MLD ASP = 28.5 ± 5.7m; MLD PIP = 24.9 ± 3.7m; p-value = 0.03 and PAR ASP = 31.5 ± 5.4 Einstein $m^{-2}$ $d^{-1}$; PAR PIP = 26.5 ± 6.7 Einstein $m^{-2}$ $d^{-1}$; p-value = 0.02).

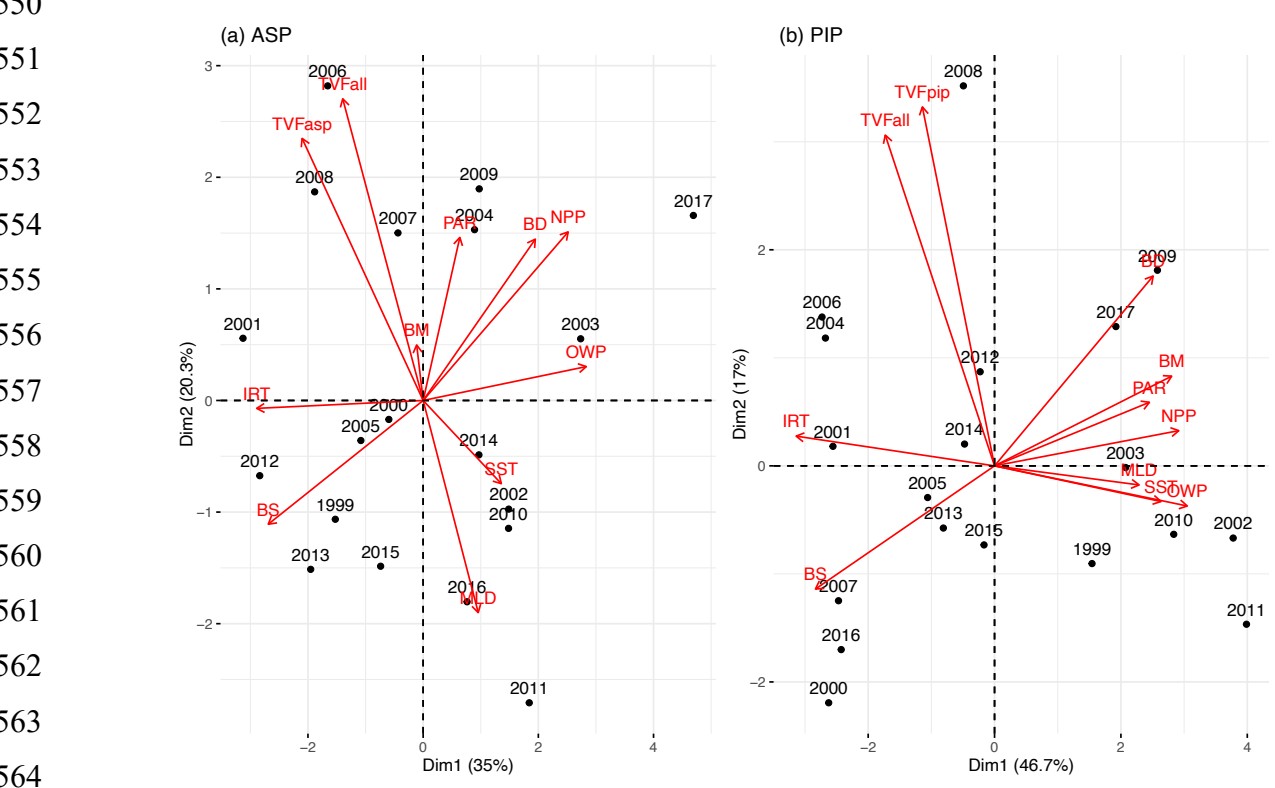

**Fig. 8.** Principal component analysis biplot of environmental parameters (red) and years (black) for (a) the Amundsen (ASP) and (b) the Pine Island (PIP) polynyas. TVFasp = total volume flux for ASP; TVFpip = total volume flux for PIP; TVFall = total volume flux for all ice shelves; BM = bloom mean; PAR = photosynthetically available radiation; BD = bloom duration; NPP = net primary productivity; OWP = open water period; SST = sea surface temperature; MLD = mixed-layer depth; BS = bloom start; IRT = ice retreat time. The same plot is presented in supplementary Fig. S6, but removing year 2001/02 for the ASP, emphasizing the relationship between total volume flux (TVFall, TVFasp) and BM in the ASP.

Finally, we find on average weak spatial negative relationships between SIC and ASL latitude, longitude, mean sector and actual central pressure in both polynyas during the growing season (Supplementary Fig. S7), and only slightly significant in the eastern PIP.

**4. Discussion**

4.1 Effect of glacial meltwater on phytoplankton chl*a* and NPP

The relationship between glacial meltwater, surface chl*a* and NPP observed over the last two decades was distinctly different between the two polynyas. In the ASP, we found that enhanced glacial meltwater translates into higher surface chl*a*, but not with NPP (when removing year 2001/02; Figs. 4a-b; Supplementary Fig. S6a). Modelling results (Fig. 6) suggest that sediment from the seafloor is the main source of dFe in the ASP, but this source is also linked to glacial melt. Ice shelf glacial meltwater drives the meltwater pump, which brings up mCDW and fine-grained subglacial sediments to the surface. This result is in agreement with previous research: Melt-laden modified CDW flowing offshore from the Dotson ice shelf to the central ASP (Sherrell et al., 2015), and resuspended sediments (Dinniman et al., 2020; St-Laurent et al., 2017; 2019) have been identified as significant sources of dFe to be used by phytoplankton. Interestingly, both dFe supplied from ice shelves and CDW are most important in front of the Thwaites and Crosson ice shelves, where the area averaged basal melt rate, and thus likely the area averaged meltwater pumping (Jourdain et al., 2017), are typically strongest in observations (Adusumilli et al., 2020; Rignot et al., 2013) and modelling (Fig. 6). The year 2001/02 does not stand out as being influenced by any specific parameter in the ASP compared to other years (Fig. 8a, Supplementary Fig. S6a). The anomalously high surface chl*a* observed during this year, as also reported by Arrigo et al. (2012), may result from exceptional conditions that were not captured by the parameters analysed in our study, for instance, an imbalance in the grazing pressure. Interestingly, surface chl*a* and NPP exhibit contrasting trends when averaged across the polynya. While TVFall may explain some of the variance in surface chl*a*, it does not account for the variance in NPP, whether assessed through direct or multivariate relationships. This decoupling between chl*a* and NPP in the ASP suggests that glacial meltwater, while enhancing surface phytoplankton biomass through nutrient delivery, may also promote vertical mixing. This mixing deepens the mixed layer, reducing light availability and constraining photosynthetic rates. These rates are influenced by fluctuations in the MLD, even in the presence of high biomass and sufficient macronutrients. The summer MLD is deeper in the ASP, which would decrease light availability, despite higher PAR compared to the PIP. Previous studies report that

the small prymnesiophyte *P. antarctica,* a low-efficiency primary producer (Lee et al., 2017), is
better adapted to deeper mixed layers and therefore lower light conditions (Alderkamp et al.,
2012; Mills et al., 2010) and could contribute to high surface chl*a* decoupled from NPP, as
observed in the ASP. This is consistent with past *in situ* studies showing systematic differences
in mixed-layer structure between the two polynyas. The PIP commonly exhibits a shallow,
strongly stratified surface mixed layer while the ASP is more variable and has been observed to
host deeper MLD (Alderkamp et al., 2012; Park et al., 2017; Yager et al., 2016; Mills et al.,
2012). Near glacier and ice-shelf fronts, entrainment of iron-rich deep waters rising to the surface
through the meltwater pump can also produce surface chl*a* maxima (high biomass from. *P.*
*antarctica*) without proportional increases in depth-integrated productivity due to self-shading
(Twelves et al., 2021). Further from the coast, meltwater spreading at neutral buoyancy
strengthens stratification, limiting vertical nutrient fluxes and thereby suppressing NPP despite
elevated chl*a*. These dual mechanisms are consistent with observational and modelling studies of
meltwater entrainment and dispersal (Randall-Goodwin et al., 2015; St-Laurent et al., 2017;
Dinniman et al., 2020; Forsch et al. 2021), and suggest that spatial heterogeneity in plume
dynamics could explain the observed chl*a* and NPP mismatch.

In the PIP, we did not find any long-term relationships between the phytoplankton bloom, NPP
and glacial meltwater. Variability in ice shelf glacial meltwater may not have the same effect on
the surface chl*a* and NPP in the PIP compared to the ASP. Iron delivered from glacial melt
process related in the PIP and west of it could accumulate and follow the westward coastal
current, towards the ASP (St-Laurent et al., 2017). These sources would include dFe from
meltwater pumped CDW, sediments and ice shelves, all of which are higher in front of the
Crosson ice shelf, west of the PIP (Fig. 6). With the coastal circulation, this would make dFe
supplied by glacial meltwater greater in the ASP, thereby contributing to the higher productivity
in the ASP. Recently, subglacial discharge (SGD) was shown to have a different impact on basal
melt rate in the ASE polynyas (Goldberg et al., 2023), where PIG had a lot less relative increase
in melt with SGD input than Thwaites or Dotson/Crosson. Thus, assuming a direct relationship
between meltrate, SGD and dFe sources, the signal in the PIP (fed by PIG melt) will be much
weaker than in the ASP (fed by upstream Thwaites, Crosson and local Dotson due to the
circulation), which might also explain the discrepancies between the PIP and ASP. A stronger
meltwater-driven stratification may also dominate in the PIP, reducing vertical nutrient
replenishment and thereby limiting biomass growth (Oh et al., 2022), even where TVFall is high,
hence leading to a direct negative relationship observed compared to the ASP (Fig. 4;
Supplementary Fig. S4). The model outputs used here are critical to understand the spatial
distribution of dFe in the embayment. They strongly suggest, but do not definitively demonstrate,
the role of dFe in influencing the phytoplankton bloom interannual variability.

Satellite algorithms commonly estimate NPP from surface chl*a*, but the approach and
assumptions vary across models. The VGPM relates chl*a* to depth-integrated photosynthesis
through empirical relationships with light and temperature (Behrenfeld & Falkowski, 1997). In
contrast, the Carbon-based Productivity Model (CbPM) emphasizes phytoplankton carbon
biomass and growth rates derived from satellite optical properties, decoupling productivity
estimates from chl*a* alone (Westberry et al., 2008). The CAFE model (Carbon, Absorption, and
Fluorescence Euphotic-resolving model) integrates additional physiological parameters such as
chl*a* fluorescence and absorption to better constrain phytoplankton carbon fixation (Silsbe et al.,
2016). In the Southern Ocean, where light limitation, iron supply, and community composition
strongly influence the relationship between chl*a* and productivity, these algorithmic differences
can yield substantial variability in NPP estimates (Ryan-Keogh et al., 2023), with studies
showing that VGPM-type models often outperform CbPM in coastal Southern Ocean regions
(Jena et al., 2020). Because the VGPM algorithm does not explicitly incorporate the MLD, but
instead estimates primary production integrated over the euphotic zone based on surface chl*a*,
PAR, and temperature, it may not fully capture the influence of variable MLD or subsurface
processes related to glacial melt, which could contribute to the observed decoupling between
chl*a* and NPP. Therefore, while the observed decoupling between chl*a* and NPP in the ASP
might also come from our choice of dataset, the VGPM model may be more appropriate for
coastal polynya environments, such as those in the Amundsen Sea. We finally note as a
limitation that satellite-derived chl*a* and VGPM NPP estimates lack the vertical resolution
needed to resolve sub-plume stratification and mixing processes (e.g., fine-scale vertical
gradients in iron or nutrient fluxes), so our mechanistic interpretations of surface chl*a* vs. depth-
integrated productivity decoupling must be taken with caution.

Direct observations from Sherrell et al. (2015) showed higher chl*a* in the central ASP while
surface dFe was low weeks before the bloom peak. This suggests a continuous supply and
consumption of dFe in the area, most likely from the circulation, as mentioned earlier.
Considering the long residence time of water masses in both polynyas (about 2 years (Tamsitt et
al., 2021)), and the daily dFe uptake by phytoplankton (3-196 pmol $l^{-1}$ $d^{-1}$ (Lannuzel et al.,
2023)), we also hypothesise that any dFe reaching the upper ocean from external sources is
quickly used and unlikely to remain readily available for phytoplankton in the following spring
season.

In recent model simulations with the meltwater pump turned off, Fe becomes the principal factor
limiting phytoplankton growth in the ASP (Oliver et al., 2019). However, the transport of Fe-rich
glacial meltwater outside the ice shelf cavities and to the ocean surface depends strongly on the
local hydrography. While Naveira Garabato et al. (2017) suggested that the glacial meltwater
concentration and settling depth (neutral buoyancy) outside the ice shelf cavities is controlled by
an overturning circulation driven by instability, others suggest that the strong stratification plays
an important role in how close to the surface the buoyant plume of said meltwater can rise
(Arnscheidt et al., 2021; Zheng et al., 2021). Therefore, high melting years and greater TVFall
might not necessarily translate into a more iron-enriched meltwater delivered to the surface
outside the ice shelf cavities, close to the ice shelf edge, as rising water masses may be either
prevented from doing so, or be transported further offshore in the polynyas where the
phytoplankton bloom occurs, before they can resurface (Herraiz-Borreguero et al., 2016).

Although several Fe sources can fuel polynya blooms, and they depend on processes mentioned
above, Fe-binding ligands may ultimately set the limit on how much of this dFe stays dissolved
in the surface waters (Gledhill and Buck, 2012; Hassler et al., 2019; Tagliabue et al., 2019).
Models of the Amundsen Sea (Dinniman et al., 2020, 2023; St-Laurent et al., 2017, 2019) did not
include Fe complexation with ligands and assumed a continuous supply of available dFe for
phytoplankton. Spatial and seasonal data on Fe-binding ligands along the Antarctic coast remain
extremely scarce and their dynamics are poorly understood (see Smith et al. (2022) for a
database of publicly available Fe-binding ligand surveys performed south of 50°S). Field
observations in the ASP and PIP suggest that the ligands measured in the upwelling region in
front of the ice shelves had little capacity to complex any additional Fe supplied from glacial
melt. As a consequence, much of the glacial and sedimentary Fe supply in front of the ice
shelves could be lost via particle scavenging and precipitation (Thuróczy et al., 2012). This was
also observed by van Manen et al. (2022) in the ASP. However, within the polynya blooms,
Thuróczy et al. (2012) found that the ligands produced by biological activity were capable of
stabilising additional Fe supplied from glacial melt, where we observed the highest productivity.
The production of ligands by phytoplankton would increase the stock of bioavailable dFe and
further fuel the phytoplankton bloom in the polynyas, potentially highlighting the dominance
of *P. antarctica*, which uses iron-binding ligands more efficiently than diatoms (Thuróczy et al.,
2012), even under low light conditions. Model development and sustained field observations on
dFe availability, including ligands, are needed to adequately predict how these may impact
biological productivity under changing glacial and oceanic conditions, now and in the future.

Overall, the discrepancies observed between the ASP and PIP point to a complex set of ice-
ocean-sediment interactions, where several co-occurring processes and differences in
hydrographic properties of the water column influence dFe supply and consequent primary
productivity.

4.2 Possible drivers of the difference in phytoplankton surface chl*a* and NPP between the

two polynyas

The biological productivity is higher in the ASP than the PIP, consistent with previous studies
(Arrigo et al., 2012; Park et al., 2017). In section 4.1, we mentioned the suspected underlying
hydrographic drivers of these differences. We related the higher biological productivity in the
ASP to a potentially greater supply of iron from melt-laden dFe-enriched mCDW and sediment
sources, but this difference in productivity could also be attributed to other local features. The
Bear Ridge grounded icebergs on the ASP's eastern side (Bett et al., 2020) could add to the
overall meltwater pump strength. They can enhance warm CDW intrusions to the ice shelf cavity
(Bett et al., 2020), increasing ice shelf melting and subsequent stronger phytoplankton bloom
from the meltwater pump activity. These processes are weaker or absent in the PIP. Few sources
other than glacial meltwater may influence the bloom in the PIP. For instance, dFe in the
euphotic zone can also be sustained by the biological recycling, as shown in the PIP by Gerringa
et al. (2020).

Sea ice could also partly explain the difference in chl*a* magnitudes, NPP, and variability between
the ASP and PIP. The strong spatial correlation between SIC and meridional winds (Fig. 7a)
indicates that southerly winds can export the coastal sea ice offshore and play a significant role
in opening the polynyas. In the ASP compared to the PIP, sea ice retreats earlier (IRT = Jan 1st ±
14d vs Jan 18th ± 17d, p-value = 0.003), the open water period is longer (OWP = 61 ± 16d vs 44
± 22d, p-value < 0.001), and the SIC is lower (Supplementary Fig. S8). In the ASP, an early sea-
ice retreat leads to an earlier bloom start, but the longer open water period is not significantly
associated with greater bloom mean or maximum chl*a* (Table 2). On the other hand in the PIP,
an early sea-ice retreat also triggers an early bloom start, but the longer open duration is
associated with warmer water, higher bloom mean chl*a*, maximum chl*a*, and NPP. These results
suggest that different processes might drive phytoplankton growth variability in the two
polynyas. In the ASP, it is likely the replenishment of dFe mentioned above that mostly
influences the bloom. In the PIP, higher SIC can delay the retreat time and shorten the open
water season (Table 2, Supplementary Fig. S8), leading to lower chl*a* and NPP compared to the
ASP. The significant negative relationships between IRT, PAR, chl*a* and NPP in the PIP (Table
2, Supplementary Fig. S6) suggests a strong light limitation relief in the polynya. This light
limitation hypothesis is further supported by the high correlation between polynya-averaged chl*a*
mean with PAR and SST in the PIP across the 19 years of study, compared to the lack of
correlation in the ASP (Supplementary Table T2; p-value < 0.01 for all relationships in the PIP).
While *P. antarctica* is usually the main phytoplankton species dominating in both polynyas, the
combination of light-limitation relief and higher SST may create better conditions for a stratified
and warmer environment that would favor diatom (Arrigo et al., 1999; van Leeuwe et al., 2020),
as recently observed in the ASP (Lee et al., 2022). The positive association of PAR, SST and
chl*a* with MLD likely reflects conditions around sea-ice retreat (all negatively associated with
IRT), when enhanced wind mixing deepens the mixed layer and replenishes surface nutrients
while light availability and SST increases. This nutrient-light co-limitation phase supports high
biomass accumulation, likely from diatoms. Similar results have been reported by Park et al.
(2017). They found that the PIP was dFe was not limited by dFe, potentially from biological
recycling (Gerringa et al., 2020), compared to an iron-limited ASP. We hypothesise that the
connection between glacial meltwater and chl*a* that we found in the ASP is a response to iron
input (also observed by Park et al. (2017) during incubation experiments) compared to the PIP,
where light and temperature seem to play a more significant role in driving the phytoplankton
bloom variability. Hayward et al. (2025) showed a decline in diatoms from 1997 to 2017 in the
PIP. However, they observed an increase in diatoms after 2017, linked to regime shift in sea ice.
Their study also indicates that diatoms are competitively disadvantaged under iron-depleted
conditions. *P. antarctica*, which relies on dFe supplied by ocean circulation, would then tend to
dominate in the ASP. Such shifts in phytoplankton composition are likely to affect carbon
export, grazing, and higher trophic levels. Additional long-term data on inter-annual variability
in phytoplankton composition and physiology will be essential to fully understand these
relationships.

Finally, the weak relationships between the ASL indices and SIC might be owing to the seasonal
variation of the ASL, where its position largely varies during summer, and its impact in shaping
coastal sea ice is also greater during winter and autumn in the Amundsen-Bellingshausen region
(Hosking et al., 2013). The lack of strong significant relationships overall does not allow us to
conclude that the ASL plays an important role in shaping the coastal polynyas landscape and
influencing chl*a* variability.

4.3 Limitations and future directions


We acknowledge that elevated concentrations of suspended sediments (and non-
photosynthetically active particles in general) near the ocean surface can impart optical
signatures that bias satellite-derived chl*a* high in coastal waters. Consequently, the higher chl*a*
observed in the ASP relative to the PIP, as well as the weak correspondence between chl*a* and
NPP in ASP, may reflect some sediment-driven optical effects rather than enhanced
phytoplankton biomass or productivity alone. While our results are consistent with known
differences in iron supply and mixed-layer dynamics between the two polynyas, the potential
contribution of sediment-related bias cannot be ruled out and should be acknowledged when
interpreting spatial contrasts in satellite chl*a* on the Antarctic shelf.

While it seems reasonable that the higher ASP productivity could be driven by more iron
delivered through a stronger meltwater pump downstream of the PIP, our data cannot confirm
this hypothesis. To accurately understand the role of iron through the meltwater pump process,
we would need to quantify the fraction of meltwater and glacial modified water (mix of CDW
and ice shelf meltwater) reaching the ocean surface, together with the iron content. Obtaining
this information is challenging over the decadal time scales considered and the method used in
our study. Here, our intention was to provide valuable insights into the potential drivers of our
results, and highlight the benefit of remote sensing, in this poorly observed environment. Our
work directly aligns with Pan et al. (2025), who investigated the long-term relationship between
sea surface glacial meltwater and satellite surface chl*a* in the Western Antarctic Peninsula, and
found a strong relationship between the two parameters, highlighting the importance of glacial
meltwater discharge in regions prone to extreme and rapid climate changes.

In multimodel climate change simulations, Naughten et al (2018) showed an increase of ice
shelves melting up to 90% on average, attributed to more warm CDW on the shelf, due to
atmospherically driven changes in local sea-ice formation. More recently, Dinniman et al. (2023)
also highlighted the impact of projected atmospheric changes on Antarctic ice sheet melt. They
showed that strengthening winds, increasing precipitation and warmer atmospheric temperatures
will increase heat advection onto the continental shelf, ultimately increasing basal melt rate by
83% by 2100. Compared to present climate simulations, their simulation showed a 62% increase
in total dFe supply to shelf surface waters, while basal melt driven overturning Fe supply
increased by 48%. The ice shelf melt and overturning contributions varied spatially, increasing in
the Amundsen-Bellingshausen area and decreasing in East Antarctica. This implies that, under
future climate change, phytoplankton productivity could show stronger spatial asymmetry
around Antarctica. The increasing melting and thinning of ice shelves will eventually result in
more numerous calving events and drifting icebergs (Liu et al., 2015). Model simulations
stressed the importance of ice shelves and icebergs in delivering dFe to the SO (Death et al.,
2014; Person et al., 2019), increasing offshore productivity. As Fe will likely be replenished and
sufficient from increasing melting in coastal areas, it is possible that the system will shift from
Fe-limited to being limited by nitrate, silicate, or even manganese (Anugerahanti and Tagliabue,
2024), while offshore SO productivity will likely remain Fe-dependent (Oh et al., 2022).

**5. Conclusions**

Using spatial and multivariate approaches, our study explored the variability of surface chl*a* and
NPP in the Amundsen Sea polynyas over the last two decades, with a focus on the main
environmental characteristics of the ASE. We found a strong relationship between ice shelf
melting and surface chl*a* in the ASP when year 2001/02 was removed, a result in agreement with
the ASPIRE field studies and previous satellite analyses. On the other hand, we did not find clear
evidence of such a relationship in the PIP, where light, sea surface temperature and open water
availability seem more important. The differences between the polynyas may lie in hydrographic
properties, or the use of satellite remote sensing itself, which cannot tell us about processes such
as Fe supply, bioavailability and phytoplankton demand. To gain greater insight, we referred to
model simulations that showed the spatial variability in the magnitude of iron sources. Our
results call for sustained *in situ* observations (e.g., moorings equipped with trace-metal clean
samplers, and physical sensors to better understand year-to-year water mass meltwater fraction
and properties) to elucidate these long-term relationships. Satellite observations are a powerful
tool to investigate the relationship between glacial meltwater and biological productivity on such
time scales, which has until now relied almost exclusively on field observations and modelling.
Using such tools, we showed how the relationship between phytoplankton and the environment
varies spatially and temporally across 19 years.

**Appendices**
No appendices are related to the manuscript.

**Data availability**
Bathymetry data (Amante & Eakins, 2009) was taken from the NOAA website
(http://www.ngdc.noaa.gov/mgg/global/global.html). Mixed-layer depth (ECCO Consortium et
al., 2021) can be accessed here:
https://podaac.jpl.nasa.gov/dataset/ECCO_L4_MIXED_LAYER_DEPTH_05DEG_MONTHLY
_V4R4. Satellite surface chlorophyll-*a* and photosynthetically available radiation were
downloaded from https://www.globcolour.info/. Sea surface temperature (Huang et al., 2021)
can be found here https://psl.noaa.gov/data/gridded/data.noaa.oisst.v2.highres.html. Wind re-
analysis data (Hersbach et al., 2020) are available at
https://cds.climate.copernicus.eu/datasets/reanalysis-era5-single-levels-monthly-
means?tab=download. Sea-ice concentration (Cavalieri et al., 1996) was obtained from
https://nsidc.org/data and Net Primary productivity (Behrenfeld and Falkowski, 1997) was
downloaded from http://sites.science.oregonstate.edu/ocean.productivity/index.php. Circumpolar
surface model output from Dinniman et al (2020) can be found at https://www.bco-
dmo.org/dataset/782848. The Amundsen Sea Low index (Hosking et al., 2016) data are available
at http://scotthosking.com/asl_index.

**Author contributions**
GL conceptualised and led the study; MSD provided the dissolved iron model output. All authors
were involved in the interpretation of the results, the revision, and the writing of the final version
of the paper.

**Competing interest**
We declare having no competing interests.

**Acknowledgments**
We would like to thank the University of Tasmania, the Australian Research Council (ARC)
Centre of Excellence for Climate Extremes (CE170100023), and the Australian Centre for
Excellence in Antarctic Science (ACEAS; SR200100008) for financial support. Delphine
Lannuzel is funded by the ARC through a Future Fellowship (L0026677). Sebastien Moreau
received funding from the Research Council of Norway (RCN) for the project "I-CRYME:
Impact of CRYosphere Melting on Southern Ocean Ecosystems and biogeochemical cycles"
(grant number 335512) and for the Norwegian Centre of Excellence "iC3: Center for ice,
Cryosphere, Carbon and Climate" (grand number 332635). Michael Dinniman was supported by
the U.S National Science Foundation grant OPP-1643652. We are also grateful to Will Hobbs,
Rob Massom and Patricia Yager for their knowledgeable input. We thank Vincent Georges for
some preliminary work as part of his masters' internship. We are very grateful to Fernando S.
Paolo for his early input and help with the glacial meltwater dataset. We thank the data providers
mentioned in the methods section for making their data available and free of charge.

## **Financial support**

All financial support were mentioned in the Acknowledgment section.

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
