# Peer review of "Drivers of Phytoplankton Bloom Interannual Variability in the Amundsen and Pine"

_EGUsphere, 2025_

## Author Comment (AC1)

**Response to Reviewer 1 comments**

In this document, the reviewer's comments are in black, our responses are in brown, the old original text is in light blue, the new/updated text is in dark blue.

Review of "Drivers of Phytoplankton Bloom Interannual Variability in the Amundsen and Pine Island Polynyas" by Guillaume Liniger et al.

The manuscript presents a valuable study of the phytoplankton blooms and their drivers in the Amundsen and Pine Island Polynyas. Satellite-derived Chl-a and NPP maps were used to characterize phytoplankton abundance and primary productivity in the years 1998 – 2017. Overall, the manuscript is well-written, well-organized, and the main points are clearly articulated. The determination of phenology metrics followed the standard methodologies described in the literature, and the use of Principal Component Analysis (PCA) and the Mann-Kendall test demonstrates good statistical practices. I especially appreciate the application of non-parametric statistical metrics in this study.

I see, however, a few issues that when fixed, could improve the final version of the paper. I present them in the points below:

We are grateful for the reviewer's positive feedback. We answer the reviewer's concerns below.

1. The study compares the Amundsen and Pine Island Polynyas, highlighting several differences that appear to arise from variations in local topography, sediment resuspension, and currents (as mentioned in lines 519-520). While these factors were discussed, they were overlooked in the study area's section. To improve the brief description and make it easier for readers to follow the discussion, it would be beneficial to add the following: (a) the bathymetry of the area, which is an important aspect in the analysis of sediment resuspension; (b) contrasts between the polynyas regarding circulation patterns; (c) a brief description of the glaciers analyzed in the study with a particular focus on the differences between them; and (d) information on phytoplankton composition, which determines the demand for nutrients, sensitivity to iron shortages (notably different for diatoms and small flagellates), and the potential for using recycled nutrients. Recent research has indicated changes in the phytoplankton community structure on Antarctica's shelf, including a decline in diatoms sensitive to iron shortages, so I would expect at least a brief characterization of these communities.

We thank the reviewer for their suggestions, following them, we have:

a. Added a map of the bathymetry with a schematic of ocean circulation as new Figure 1a, and the mixed-layer depth (MLD; ECCO Consortium version 4 release 4) climatology as subpanel Figure 1b to give more context to our study. We use the new MLD product later in our analysis.

**Figure 1.** Study area. Panel 1a shows the bathymetry (from ETOPO1; Amante & Eakins, 2009) and panel 1b shows the climatological summer mixed-layer depth (MLD) from 1998 to 2017. Panel 1a shows a simplified schematic of the local deep ocean circulation (~ below 400m, yellow arrows) and upper glacial meltwater/sediments/circumpolar deep water sourced dFe pathways (magenta arrows), which follows the local upper ocean circulation. Schematic adapted from St-Laurent et al. (2017). The white lines in panel (b) represent the climatological summer polynya boundaries for the Amundsen Sea polynya (left) and Pine Island polynya (right). The dark grey area is mainland Antarctica. Light grey areas indicate floating ice shelves and glaciers: Abbot (Abb), Cosgrove (Cs), Pine Island Glacier (PIG), Thwaites (Tw), Crosson (Cr), Dotson (Dt) and Getz (Gt).

b. Added a brief description of the circulation in the embayment in the method section of the updated manuscript. We also added arrows showing the local ocean circulation and pathways of glacial meltwater and sediments in the embayment in the new Figure 1a.

In the ASE, the ice sheet loss is mainly through enhanced basal melting of the ice shelves. This is attributed to an increase in wind-driven Circumpolar Deep Water (CDW) fluxes and ocean heat content intruding onto the continental shelf through deep troughs such as Pine Island and Dotson-Getz, flowing into the ice shelf cavities (Dotto et al., 2019; Jacobs et al., 2011; Pritchard et al., 2012). There, warm waters fuel intense basal melt of the Pine Island, Thwaites, and Getz ice shelves, and returns as a fresher, colder outflow that can strengthen

stratification (Jenkins et al., 2010; Ha et al., 2014). The PIP and ASP differ in their exposure to CDW and in local circulation: the ASP is more strongly influenced by upwelled modified CDW (mCDW) and glacial meltwater inputs, while the PIP is typically more stratified and less directly ventilated by deep waters (Assmann et al., 2013; Dutrieux et al., 2014). These hydrographic contrasts can shape the timing and magnitude of phytoplankton blooms and nutrient dynamics across the two polynyas."

c. Added a brief description of the relative importance of the glaciers and ice shelves in the area in the updated manuscript

We focus on the PIP and ASP in the ASE embayment in West Antarctica (Figure 1). The ASE embayment is comprised of several ice shelves and glaciers, including Abbot (Abb), Cosgrove (Cs), Pine Island (PIG), Thwaites (Tw), Crosson (Cs), Dotson (Dt) and Getz (Gt). The PIG and Thwaites have received significant attention in recent years due to their potentially large contribution to sea level rise (Rignot et al., 2019; Scambos et al., 2017). Along with the Crosson and Dotson ice shelves, the PIG and Thwaites are undergoing the highest melt rate, which is expected to increase under climate change scenarios (Naughten et al., 2023; Paolo et al., 2023). The mean MLD in the ASP is deeper (Figure 1b), indicating that it may better entrain deeper sources of nutrients into the upper waters. The polynya boundaries were determined using a 15% sea-ice concentration (SIC) mask (Moreau et al., 2015; Stammerjohn et al., 2008) for every 8-day period from June 1998 to June 2017 to accurately represent the size of the polynya through time.

d. Added information about the phytoplankton composition in the Amundsen embayment, in the introduction section in the updated manuscript.

The phytoplankton community in the ASE is generally dominated by *Phaeocystis antarctica* (Lee et al., 2012; Lee et al., 2017; Yager et al., 2016), which is adapted to low iron availability and variable light conditions, and forms large summer blooms (Alderkamp et al., 2012; Yager et al., 2016). Diatoms like *Fragilariopsis sp.* and *Chaetoceros sp.* are also present, often becoming more important near the sea-ice edge or under shallow, stratified mixed layers where silicic acid (Si) and iron (Fe) are more available (Mills et al., 2012). In exceptional years, such as 2020, diatoms like *Dactyliosolen tenuijunctus* replaced *P. antarctica* as the dominant taxon, driven by anomalously shallow mixed layers and sufficient Fe–Si supply (Lee et al., 2022). This dynamic balance highlights how light, nutrient supply, and stratification control community composition in these highly productive and complex Antarctic systems.

2. The role of ligands, which are mentioned late in the discussion, seems significant for the availability of iron to phytoplankton. Information in lines 568-570 seems to suggest a possible feedback loop between the biological activity, ligands and the bioavailability of iron, which could be an interesting aspect to consider when analyzing bloom cycles. It might be worth adding a short comment on this topic in the model description around line 190.

We agree with the reviewer. We mentioned in the original version that ligands may be important in the feedback loop "The production of ligands by phytoplankton would increase the stock of bioavailable Fe and further fuel the phytoplankton bloom in the polynyas".

Regarding how ligands are constrained in the model, in the original version of the manuscript, we specified that every iron particle in the model is bound to a ligand, implying

that all of the dissolved iron is made available at any time for phytoplankton to use. We have added a reference that showed that when dFe is bound to ligands, it remains bioavailable for phytoplankton.

"This is parameterized in the model as all iron molecules being bound to a ligand, and therefore remaining in solution in a bioavailable form (Gledhill & Buck, 2012). For a detailed and complete explanation of the model, see Dinniman et al. (2020)."

3. There are at least two GlobColour L3 chl-a products that differ by the averaging method. It would be helpful to provide an ID or DOI number for the dataset. On a similar note, it would be interesting to see a discussion on the strong connection between net primary production (NPP) and chl-a, as chl-a is a key parameter for estimating NPP.

We have specified which chla algorithm/dataset we used in section 2.2 in the updated manuscript.

"We obtained level-3 satellite surface chlorophyll-a (chla) concentration with spatial and temporal resolution of 0.04° and 8 days from the European Space Agency (ESA) Globcolor project (<a href="https://www.globcolour.info/">https://www.globcolour.info/</a>). We used the CHL1-GSM (Garver–Siegel–Maritorena) (Maritorena and Siegel, 2005) standard Case 1 water merged products consisting of the Sea-viewing Wide Field-of-view (SeaWiFS), Medium Resolution Imaging Spectrometer (MERIS), Moderate Resolution Imaging Spectroradiometer (MODIS-A) and Visible Infrared Imaging Suite sensors (VIIRS)."

We also added sentences about the relationship between chla and NPP in the method and discussion section in the updated manuscript, and elaborate more on this in #Reviewer2 response.

"The VGPM model is a chlorophyll-based approach and relies on the assumption that NPP is a function of chlorophyll, influenced by light availability and maximum daily net primary production within the euphotic zone."

"The Vertically Generalized Production Model (VGPM) relates chla to depth-integrated photosynthesis through empirical relationships with light and temperature (Behrenfeld & Falkowski, 1997)."

"We also note as a limitation that satellite-derived chla and VGPM NPP estimates lack the vertical resolution needed to resolve sub-plume stratification and mixing processes (e.g. fine-scale vertical gradients in iron or nutrient fluxes), so our interpretations of surface chla vs. depth-integrated productivity decoupling must be taken with caution."

4. Under high-nitrate low-iron conditions, literature reported significant variations in the carbon-to-chlorophyll (Cphyto:Chl) ratio from those assumed globally, due to phytoplankton adaptations to iron shortages. Additionally, low-light conditions can alter the carbon-to-chlorophyll ratio. It would be worth including these elements in the discussion as a potential source of uncertainty. Might these differences explain the significant correlation with chl-a in Figure 3 and the lack of correlation with NPP at the same time?

In the original version of the manuscript, we briefly mentioned that the decoupling between chla and NPP in the ASP could be due to the bloom being dominated by *P. antarctica*. In the updated version, we elaborate more about why we observe such differences in chla/NPP response to the environmental factors. We added some text related to the role of light and iron co-limitation, and ligands. We also added some data about the mixed-layer depth and how those could impact the phytoplankton bloom community and the relationship observed between chla, NPP and the environmental parameters throughout the text in the different section, where we hypothesize that:

a. More meltwater can drive stronger overturning, deepening the mixed layer, decreasing light availability  $\rightarrow$  Potentially favoring *P. antarctica*.

The summer MLD is deeper in the ASP (Figure 1b), which would decrease light availability, despite higher PAR compared to the PIP. Previous studies report that the small prymnesiophyte *P. antarctica*, a low-efficiency primary producer (Lee et al., 2017), is better adapted to deeper mixed layer and therefore lower light condition (Alderkamp et al., 2012; Mills et al., 2010) and could contribute to high surface chla decoupled from NPP, as observed in the ASP.

b. More ligands can favor *P. antarctica*, which can access organically bound Fe more efficiently than many diatoms.

The production of ligands by phytoplankton may increase the stock of bioavailable Fe and further fuel the phytoplankton bloom in the polynyas, potentially highlighting the dominance of *P. antarctica*, which uses iron-binding ligands more efficiently than diatoms (Thuróczy et al., 2012), even under low light conditions.

c. *P. antarctica* is adapted to a low light environment + deeper MLD, compared to diatoms. Therefore, a strong light relief + warmer temperature create an ideal stratification environment for diatom blooms to happen, which is what we suggest may be happening in the PIP based on our results.

While *P. antarctica* is usually the main phytoplankton species dominating in both polynyas, the combination of light-limitation relief and higher SST may create better conditions for a stratified and warmer environment that would favor diatoms (Arrigo et al., 1999; van Leeuwe et al., 2020), as recently observed in the ASP (Lee et al., 2022). The positive association of PAR, SST and chla with MLD likely reflects conditions around sea-ice retreat (all negatively associated with IRT), when enhanced wind mixing deepens the mixed layer and replenishes surface nutrients at the same time that light availability and SST increases. This nutrient—light co-limitation phase supports high biomass accumulation, likely from diatoms.

d. We also added some text about the potential effect of shifting community:

Our results suggest potential long-term changes in the phytoplankton community, specifically a shift towards diatoms in the ASE coastal regions during phytoplankton bloom. Hayward et al. (2025) reported a decline in diatoms from 1997 to 2017 in the PIP. However, they observed an increase in diatoms after 2017, linked to regime shift in sea ice. Their study also indicates that diatoms are competitively disadvantaged under iron-depleted conditions, whereas in the ASP, *P. antarctica*, which relies on dFe supplied by ocean circulation, tends to dominate. Such shifts in phytoplankton composition are likely to affect carbon export,

grazing, and higher trophic levels. Additional long-term data on inter-annual variability in phytoplankton composition and physiology will be essential to fully understand these relationships.

5. Lastly, a small editorial note: lines 274-277 contain a repeated sentence

Thank you for pointing it out. Corrected.

---

## Author Comment (AC2)

**Response to Reviewer 2 comments**

In this document, the reviewer's comments are in black, our responses are in brown, the old original text is in light blue, the new/updated text is in dark blue.

This manuscript uses remote sensing data products, together with some model output, to investigate whether changes in glacial melt can account for variability in surface chlorophylla and net primary productivity in two Antarctic polynyas from 1998-2017 (Amundsen Sea Polynya and Pine Island Polynya). The authors report observational support for a positive relationship between surface chla and glacial melt (but no relationship with NPP) in ASP, and no significant relationship with either variable in PIP. Instead, local processes at PIP seem to impact the bloom phenology. The authors investigate and discuss plausible mechanisms that may be operating distinctly in each region.

The authors provide a speculative discussion on why the relationship between glacial melt and chla differs between the two polynyas and why chla and NPP appear decoupled in ASP. The methods and statistical analyses are appropriate, the manuscript is logically structured, and the figures are generally insightful. Ultimately, the study points towards interesting signals and empirical results and is therefore appropriate for publication. However, there are general concerns about the mechanistic interpretation of some of the signals, as well as queries about the quality and reliability of the chla data product in this region. There are a few further specific suggestions to improve the manuscript.

We thank the reviewer for their time in reviewing our manuscript and their positive recommendation. We address their concerns below.

**General comments**

**1. Discussion of Chla and NPP relationship**

The decoupling of NPP and chla in the ASP is a central feature of the results but is not given much attention in the subsequent discussion. The explanation that it is due to the vertical mixing that may concurrently be promoted by glacial meltwater is somewhat unsatisfying. In the first instance, there is presumably a spatial separation between these processes – the meltwater plume will promote mixing at or near the glacier face, but to what extent is that enhanced mixing present across the rest of the polynya area? In contrast one might expect the meltwater to enhance stratification once it settles at a level of neutral buoyancy. Has this proposed enhancement of vertical mixing, and its spatial extent, been described elsewhere in the literature?

We thank the reviewer for pointing this out and agree that more clarification is needed.

As mentioned by the reviewer, a valid interpretation for the decoupling of chla and NPP in the ASP is the result of two processes. Near glacier and ice-shelf fronts, buoyant meltwater plumes entrain iron-rich deep water (meltwater pump), that can lead to localized surface chla maxima (i.e high biomass) without proportional increases in depth-integrated NPP. Away from the coast, meltwater spreading at neutral buoyancy can enhance stratification and can suppress nutrient supply, thereby limiting NPP despite elevated surface chla (accumulation). Observations and models support both localized entrainment and broader stratification effects (e.g., Randall-Goodwin et al., 2015; St-Laurent et al., 2017; Dinniman et al., 2020). We have

added a short paragraph to the Discussion summarizing these mechanisms and citing observational and modelling studies (e.g. Randall-Goodwin et al. 2015; St-Laurent et al. 2017; Dinniman et al. 2020; Forsch et al. 2021). We also note that the data available to us (satellite chla and VGPM NPP) cannot fully resolve vertical structure at the plume scale; we therefore frame this interpretation as a plausible mechanistic explanation consistent with previous work.

"The decoupling between surface chla and NPP could reflect two contrasting meltwater effects. Near glacier and ice-shelf fronts, entrainment of iron-rich deep waters rising to the surface through the meltwater pump can produce surface chla maxima (high biomass) without proportional increases in depth-integrated productivity. Further from the coast, meltwater spreading at neutral buoyancy strengthens stratification, limiting vertical nutrient fluxes and thereby suppressing NPP despite elevated chla. These dual mechanisms are consistent with observational and modelling studies of meltwater entrainment and dispersal (Randall-Goodwin et al., 2015; St-Laurent et al., 2017; Dinniman et al., 2020; Forsch et al. 2021) and suggest that spatial heterogeneity in plume dynamics could explain the observed chla and NPP mismatch. We also note as a limitation that satellite-derived chla and VGPM NPP estimates lack the vertical resolution needed to resolve sub-plume stratification and mixing processes (e.g. fine-scale vertical gradients in iron or nutrient fluxes), so our interpretations of surface chla vs. depth-integrated productivity decoupling must be taken with caution."

Secondly, is your explanation that deeper mixed layers limit light availability and reduce NPP relative to chla consistent with the VPGM algorithm? In what way does that algorithm take mixed layer depth into account, and is it likely to capture variations in mixed layer depth due to glacial melt in this region? This is presumably testable by looking more closely at the chla:NPP relationship directly, rather than through the lens of their relationship with TVF. The claim related to the possible role of phytoplankton community composition needs to be described in greater detail.

We thank the reviewer for highlighting this important point. The VGPM algorithm does not explicitly include mixed layer depth as an input variable (but the euphotic zone instead); NPP is estimated from surface chlorophyll, sea surface temperature, and photosynthetically available radiation (PAR) using a fixed relationship for maximum photosynthetic efficiency. As such, the VGPM algorithm may underestimate the effects of variable MLD on the light environment, particularly in regions where glacial melt modifies stratification. Our interpretation that deeper mixed layers can decouple chla from NPP should therefore be viewed as a mechanistic explanation that extends beyond what VGPM can directly resolve. We have added the MLD as a new parameter (ECCO Consortium, version 4 release 4) in our analysis that could help us explain the differences. We found that the MLD is significantly deeper on average in the ASP compared to the PIP. We found no significant correlation in the ASP and PIP between MLD and NPP. We have added text about the limitation of the VGPM, as well as the impact the MLD could have on the chla/NPP in the polynyas, with an impact on the phytoplankton community.

"The summer MLD is deeper in the ASP (Figure 1b), which would decrease light availability, despite higher PAR compared to the PIP. Previous studies report that the small prymnesiophyte *P. antarctica*, a low-efficiency primary producer (Lee et al., 2017), is better adapted to deeper mixed layers and therefore lower light conditions (Alderkamp et al., 2012; Mills et al., 2010) and could contribute to high surface chla decoupled from NPP, as observed in the ASP."

"Because the VGPM algorithm does not explicitly incorporate the MLD, but instead estimates primary production integrated over the euphotic zone based on surface chla, PAR, and temperature, it may not fully capture the influence of variable MLD or subsurface processes related to glacial melt, which could contribute to the observed decoupling between chla and NPP."

**2. Chla product and uncertainty**

Ocean colour is influenced by absorption from pure water, dissolved compounds, phytoplankton, and suspended sediments. Globally tuned chla algorithms do not always perform optimally in optically complex waters. In the context of this study, glacial meltwater could impart an optical signature potentially affecting the accuracy of chla estimates. Could the authors comment on whether the chla algorithm used is expected to handle such conditions, and how confident they are in its performance in the study region? Some additional analysis looking at the uncertainty in the chla fields or comparison with other available chla products (that use different algorithms) would be useful to gauge the potential impact of additional optical influences on the results. What influence does the number of visible days in the region have on the results? Is there reason to be confident that, in this region, a fraction of the primary production is not missed prior to the return of sufficient light for ocean colour to be detected? See a couple of recent papers that have noted possible distinctions between what the satellite sees and what growth is taking place, both with respect to the solar angle and the sea ice cover (McLish and Bushinsky, 2023; Douglas et al., 2024).

We thank the reviewer for raising this point. We agree that glacially influenced waters can be optically complex, and that globally tuned chlorophyll algorithms may have increased uncertainty in such conditions. In this study we used the GlobColour merged chlorophyll product, which has been widely applied in Southern Ocean and coastal Antarctic studies (Ardyna et al., 2017; El Dine et al., 2025; Golder & Antoine, 2025; Nunes, Fereira & Brito, 2025). We also chose the GlobColour product compared to solely MODIS-AQUA or SEAWIFS for 2 main reasons: The spatial and the temporal gaps. Using a merged product significantly increases the spatial and temporal coverage. A full inter-sensor comparison is the subject of a recent study (Garnesson et al., 2019; https://doi.org/10.5194/os-15-819-2019) and is beyond the scope of our study. Limitations of using satellite remote sensing in remote and coastal areas were also described in detail in Liniger et al. (2020). As our goal is to investigate long term relationships, we deemed the product we use more appropriate. While we cannot fully rule out bias from coloured dissolved organic matter or suspended sediments associated with meltwater, the optical complexity in Antarctic polynyas is generally lower than in Arctic, temperate or tropical coastal systems (such as estuaries or fjords, with very high CDOM concentration due to rivers discharge), and we therefore expect the product to provide a robust representation of chla variability. In the original version, we also made sure to explicitly state that we focused on surface chla and NPP and that some productivity could be missed, we have added the suggested references by the reviewer, as well as Stoer & Fennel 2024, to strengthen this point.

"We caution that our study focuses on surface productivity, and satellites cannot detect under-ice phytoplankton and sea-ice algal blooms, therefore likely underestimating total primary productivity (Ardyna et al., 2020; Boles et al., 2020; Douglas et al., 2024; McClish & Bushinsky, 2023; Stoer & Fennel., 2024)."

We have added extensive text in the updated manuscript about (1) our choice of selecting the GlobColour dataset in the Method section, (2) the limitation of surface chla algorithms and the lack of long term in situ data to perform strong match-ups and (3) why we believe VGPM is well adapted for our area of interest.

"We chose to perform our analysis with the merged GlobColour product to increase our spatial and temporal coverage".

"We note that satellite ocean-colour chla algorithms (including the GlobColour merged product used here) are globally tuned and may underperform in optically complex waters (e.g., with elevated dissolved organic matter or suspended sediments, 'Case 2'). In the Amundsen Sea Polynya, past work (Park et al. 2017) shows that satellite chla climatologies reflect broad seasonal patterns that are consistent with *in situ* measurements of phytoplankton biomass and photophysiology, but there is limited data from regions immediately adjacent to glacier fronts or during times of strong meltwater input. Thus, while we consider satellite chla to be useful for capturing spatial and temporal variability at polynya scale, uncertainty likely increases in optically complex zones near glacier margins or during low-light periods and needs to be considered while interpreting results."

"Satellite algorithms commonly estimate NPP from surface chla, but the approach and assumptions vary across models. The Vertically Generalized Production Model (VGPM) relates chla to depth-integrated photosynthesis through empirical relationships with light and temperature (Behrenfeld & Falkowski, 1997). In contrast, the Carbon-based Productivity Model (CbPM) emphasizes phytoplankton carbon biomass and growth rates derived from satellite optical properties, decoupling productivity estimates from chla alone (Westberry et al., 2008). The CAFE model (Carbon, Absorption, and Fluorescence Euphotic-resolving model) integrates additional physiological parameters such as chlorophyll fluorescence and absorption to better constrain phytoplankton carbon fixation (Silsbe et al., 2016). In the Southern Ocean, where light limitation, iron supply, and community composition strongly influence the relationship between chla and productivity, these algorithmic differences can yield substantial variability in NPP estimates (Arrigo et al., 2008). Consequently, the choice of algorithm strongly influences NPP estimates (Ryan-Keogh et al., 2023), with studies showing that VGPM-type models often outperform CbPM in coastal Southern Ocean regions (Jena et al., 2020). Therefore, while the observed decoupling between chla and NPP in the ASP might come from the choice of dataset, the VGPM model may be more appropriate for coastal polynya environments, such as those in the Amundsen Sea.".

**3. Oceanographic context**

The introduction could be strengthened by providing some background on the regional circulation and major water masses influencing the study area and how this differs between the two polynyas. Similarly, the description of the meltwater pump was somewhat lacking in detail. Expanding this section would help better frame and qualify the later discussion.

We have updated our new Figure 1, representing the full embayment with the bathymetry, arrows showing the circulation (Fig1a), and the summer average MLD (Fig1b). We also added some text about the circulation and the water masses.

"The ASE is also the Antarctic region experiencing the highest mass loss from the Antarctic ice sheet. It has been undergoing increased calving, melting, thinning and retreat over the past three decades (Paolo et al., 2015; Rignot et al., 2013; Rignot et al., 2019; Shepherd et al.,

2018). In the ASE, this ice sheet loss is mainly through enhanced basal melting of the ice shelves. This is attributed to an increase in wind-driven Circumpolar Deep Water (CDW) fluxes and ocean heat content intruding onto the continental shelf through deep troughs such as Pine Island and Dotson-Getz, flowing into the ice shelf cavities (Dotto et al., 2019; Jacobs et al., 2011; Pritchard et al., 2012). There, warm waters fuel intense basal melt of the Pine Island, Thwaites, and Getz ice shelves, and returns as a fresher, colder outflow that can strengthen stratification (Jenkins et al., 2010; Ha et al., 2014). The PIP and ASP differ in their exposure to CDW and in local circulation: the ASP is more strongly influenced by upwelled modified CDW (mCDW) and associated meltwater inputs, whereas in the PIP, CDWvertical intrusions primarily occur beneath the ice shelves, leading to a more stratified and less directly ventilated surface layer (Assmann et al., 2013; Dutrieux et al., 2014). These hydrographic contrasts can shape the timing and magnitude of phytoplankton blooms and nutrient dynamics across the two polynyas."

Regarding the meltwater pump, we believe that our original description sums up the processes concisely, that being:

"Melting ice shelves can explain about 60% of the phytoplankton biomass variance between all Antarctic polynyas, suggesting that they are the primary supplier of dissolved iron (dFe) to coastal polynyas (Arrigo et al., 2015), and can directly or indirectly contribute to regional

marine productivity (Bhatia et al., 2013; Gerringa et al., 2012; Hawkings et al., 2014; Herraiz-Borreguero et al., 2016). The strong melting of the ice shelves can release significant quantities of freshwater at depth (Biddle et al., 2017), resulting in a strong overturning within the ice shelf cavity, called the meltwater pump (St-Laurent et al., 2017). Modelling efforts have identified both resuspended Fe-enriched sediments and CDW entrained to the surface by the meltwater pump as the two primary sources of dFe to coastal polynyas, providing up to 31% of the total dFe, compared to 6% for direct ice shelf input (Dinniman et al., 2020; St-Laurent et al., 2017)."

Additionally, there is some confusing use of terminology that could be clarified. In particular, "ice shelf meltwater" and "glacial meltwater" seem to be used interchangeably throughout the manuscript, while "subglacial discharge" is used only once in the discussion but is not defined anywhere. Suggest using consistent terminology and provide a clear distinction between terms.

Thank you for picking this up. We have modified our terminology and decided to keep 'glacial meltwater' and specified this the end of section 2.3 what the term defines:

"We use the term glacial meltwater which defines meltwater resulting from ice shelf melting"

**4. Figure presentation**

Many of the maps are too small, leading to overlapping of labels, and obscuring of data with overlaid markings. Please make maps larger, especially for Figures 4 and 6.

Thank you for pointing this out, we have updated all figures to make them bigger and clearer.

5. Data availability and reproducibility

Please make sure to provide all details needed to locate and access the versions of the data products used, rather than simply links to the general websites. DOI's should be provided where available.

We have added the necessary information including references to the datasets and DOI when available, in the acknowledgment and the reference section.

**Specific comments**

L79: what does "natural" mean in this context?

We meant from natural sources, which is confusing. This is a mistake from our end and the word was removed. Thank you.

L147-153: Given the NPP dataset is central to the main results of the manuscript, I suggest including a few lines on how NPP was derived in the model. This would also be useful for the later discussion.

We have added the description of how NPP is calculated from the VGPM model in the method section, as well as in the discussion when discussing other algorithms.

Method: "The VGPM model is a chlorophyll-based approach and relies on the assumption that NPP is a function of chlorophyll, influenced by light availability and maximum daily net primary production within the euphotic zone."

Discussion: "The Vertically Generalized Production Model (VGPM) relates chla to depth-integrated photosynthesis through empirical relationships with light and temperature (Behrenfeld & Falkowski, 1997)."

L274: Why highlight the relationship between chla in ASP and TVFasp and Dotson ice shelf but not Crosson ice shelf?

This was an omission from our end; We have added the Crosson ice shelf in the sentence.

"On the other hand, ASP chla shows strong relationships with TVFasp, the Dotson and Crosson ice shelves (Table 1)."

L304-343: Table 1 and Figure 4 show opposing relationships between chla and TVF for ASP and PIP. While the relationship is positive for ASP it is negative for PIP (particularly for Cosgrove where the relationship is quite strong). How do you interpret this difference? The ASP is more directly ventilated by intrusions of CDW that interact with glaciers (becoming mCDW) and upwell near the surface. The resulting meltwater is moderate and iron-rich, enhancing and overturning circulation, bringing nutrients back to the surface, followed by stratification which fuels the phytoplankton growth where nutrients are abundant. Hence, a positive relationship between chla and TVF. The PIP generally experiences stronger and more persistent stratification and is less ventilated by CDW. When meltwater input is high, the surface layer can become nutrient-depleted despite remaining strongly stratified, which leads to lower NPP and chla, even as TVF increases. Thus, this could explain a negative chla/TVF relationship. However, because this relationship is not statistically significant, we decided to not pursue its investigation and solely focused on the ASP relationships.

We have added text in the discussion mentioning this:

"A stronger meltwater-driven stratification may also dominate in the PIP, reducing vertical nutrient replenishment and thereby limiting biomass growth (Oh et al., 2022), even where TVF is high, hence leading to a direct negative relationship observed compared to the ASP (Figure 4 and Supplementary Figure S4), where mixing is promoted through the meltwater pump."

**L333. Comments for Figure 4;**

- Suggest using black crossing to indicate insignificant correlations rather than significant ones. At present, it is difficult to read the magnitude of the correlations because the colours are obscured by the black crosses.

Thank you for your suggestion. Instead of marking the non-significant relationship with black crosses, we have opted to still highlight the significant relationships using a contour of 0.05 (p-value significance). Please see the updated figures.

- Consider repeating the labels from Figure 1 to guide the reader (at least for the ice shelves) All figures have been updated with the ice shelves labels.
- Ensure longitudinal labels are legible and not overlapping Done.

L421-422: "IRT and OWP are significantly related in the PIP." Is this also true for ASP? Where is this relationship shown?

We have added the statistics for both relationships in the updated text.

"IRT and OWP are significantly related (rho = -0.93; p-value < 0.001)." "IRT and OWP are significantly related in the PIP (rho = -0.88; p-value < 0.001)."

L443-444: Did you do any pretreatment of the data, e.g. mean centering and normalisation? Please specify or alternatively, argue for why you did not do this.

We did not apply mean-centering or normalization to the variables before performing PCA. The variables are already expressed in comparable physical units and have similar ranges, so scaling or centering is not strictly required. This approach is consistent with other studies in marine biogeochemistry that perform PCA directly on raw environmental data (Marchese et al., 2017; Liniger et al., 2020). Furthermore, Reid & Spencer (2009, <a href="https://doi.org/10.1016/j.envpol.2009.03.033">https://doi.org/10.1016/j.envpol.2009.03.033</a>) examined the influence of various data pretreatment methods on PCA outcomes in estuarine and coastal water quality studies. The study

found that while pretreatment can be beneficial in certain contexts, they are not always necessary, particularly when variables are expressed in comparable units and ranges. We have now clarified this choice in the Methods section.

"No pre-treatment (mean-centering or normalization) was applied to the variables prior to PCA, as all variables are expressed in comparable units and ranges, consistent with common practice in marine biogeochemistry studies (Marchese et al., 2017; Liniger et al., 2020)"

L454-456: The loadings (vectors) for OWP and IRT are very similar in ASP and PIP, both in their projections onto Dim1 and Dim2 and in their magnitudes. The main difference between the two polynyas with regards to OWP and IRT lies in the variance explained by Dim1. I suggest using this difference to support the statement that "...physical conditions might play a stronger structuring role..." rather than how they project on Dim1 and Dim2. We have followed the reviewer suggestions and added a statement about Dim1 in the updated manuscript:

"Compared to the ASP, both NPP and BM clustered strongly with BD, and PAR. Additionally, IRT, OWP and SST and MLD aligned along Dim1, which explains 46.7% of the total variance compared to 35% for the ASP, suggesting that physical conditions might play a stronger structuring role in PIP compared to the ASP."

L457-458: which is in line with the earlier correlation analysis showing opposing relationships between chla and TFV between the two polynyas.

L476: Comment for Figure 7:

General: Please explain in more detail in the main text how to interpret this figure, and PCAs in general, for the uninitiated.

We added text at the beginning of the paragraph to help the reader with the comprehension of the plot.

"The PCA reduces our datasets (11 variables) and breaks them down into dimensions that capture most of the variability and relationships between all variables. Arrows indicate the contribution of each variable to the dimensions, with longer arrows representing stronger influence. Observations (in our case, years) positioned in the direction of an arrow have a stronger influence of that variable".

- Figure 7 and the accompanying text heavily relies on the use of acronyms. You might consider providing a legend next to panel b for ease of interpretation. We have followed the reviewers' suggestion and have explicitly described each acronym in the figure caption. Please see the new figure caption.
- Please ensure labels are not overlapping

The labels that appear on the biplot are automatically added when the PCA plot is generated. We understand this can somehow impede the reading but believe Figure 8 reads well overall.

L546: "settling depth" is unclear. Do you mean the depth of neutral buoyancy? Yes, that is what we meant, we have specified it in the updated manuscript.

"While Naveira Garabato et al. (2017) suggested that the glacial meltwater concentration and settling depth (neutral buoyancy) outside the ice shelf cavities is controlled by an overturning circulation driven by instability."

**Minor/technical comments**

L50: Reference Figure 1.

We have updated Figure 1 and believe the reference of this line to the new figure is not appropriate anymore. Instead, we refer Figure 1 later when we give a description of the embayment circulation in section 2.1

L276-277: Delete duplicate sentence.

Done.

L278-279: This has already been stated. Delete.

Done.

L446: Please make sure you define all acronyms. "BD" is currently not defined.

L454: As above, please define "BM".

Thank you for pointing this out. We have added the acronyms in the Methods section when describing the bloom phenology metrics, and have described all acronyms for Figure 8 in the figure caption.

L498: change "... and the modelling..." to and models.

L519: delete "related".

Done.

Done.

---

## Author Response (AR2)

**Response to Reviews**

In this document the editor's and reviewers' comments are in black, our responses are in brown, and the amended or new text is in blue

- - - - - - - - - - - - - - - - - - - - - - - - - - - - - - -

Dear Dr. Guillaume Liniger,

We have received additional comments from the second round of reviews, which are appended below. One remaining major concern is the potential bias in the Chl-a data, and this needs to be addressed before further consideration for publication. I am therefore returning the manuscript to you so that you can make the necessary revisions.

Best regards,
Yuan Shen
Associate Editor

We thank the editor for this comment and request. We address all remaining concerns below.

- - - - - - - - - - - - - - - - - - - - - - - - - - - - - - -

Review notes for
*Liniger et al., Drivers of phytoplankton bloom interannual variability in the Amundsen and Pine Island Polynyas*

Firstly, we apologies for the delay in returning our review.
We thank the authors for preparing the revised manuscript and for their efforts in addressing the comments from both Reviewer #1 and ourselves. Many of our concerns have been satisfactorily resolved, however, we remain somewhat unsatisfied with the response to our concern regarding possible bias in the chl-a data product (see general comment below). In addition, we provide a few very minor comments. Once these issues are resolved, we believe the manuscript would be suitable for publication.

We thank both reviewers for taking the time to review our manuscript once again. We address their final concerns below.

**General comments**
1) Chl-a data product:
While we appreciate the additional details provided in the revised manuscript, the authors have not addressed the central issue, namely that sediments may impart an optical signature in surface waters that may introduce bias in the Chl-a data product. This concern cannot be dismissed by noting that previous studies have used the same

product. The primary explanation offered for the differing chla–meltwater relationships between ASP and PIP is that ASP is more strongly influenced by sediments. Following the same logic, a stronger influence of sediments in ASP could artificially elevate the retrieved chla relative to PIP not because of alleviation of iron limitation and stimulation of phytoplankton productivity, but due to bias in the chla data product. This would also provide an alternative explanation for the decoupling between chla and NPP in ASP. We understand that an uncertainty analysis is beyond the scope of this manuscript. However, because a possible influence of sediments cannot be ruled out, we request that this be explicitly acknowledged in the limitations section.

In the previous version of the manuscript, we added the following text regarding potential biases in our study, L185-194:
*"We note that satellite ocean-colour chla algorithms (including the GlobColour merged product used here) are globally tuned and may underperform in optically complex waters (e.g., with elevated dissolved organic matter or suspended sediments, 'Case 2'). In the ASP, past work (e.g., Park et al. 2017) shows that satellite chlorophyll climatologies reflect broad seasonal patterns that are consistent with in situ measurements of phytoplankton biomass and photophysiology, but there is limited data from regions immediately adjacent to glacier fronts or during times of strong meltwater input. Thus, while we consider satellite chla to be useful for capturing spatial and temporal variability at polynya scale, uncertainty likely increases in optically complex zones near glacier margins or during low-light periods, and needs to be considered while interpreting results."*.

In the updated version, we added more text based on the reviewer's comment regarding the influence of sediment and how it could impact the chla estimates, as well as the chla-npp relationship.

**New text:** "We acknowledge that elevated concentrations of suspended sediments (and non-photosynthetically active particles in general) near the ocean surface can impart optical signatures that bias satellite-derived chla high in coastal waters. Consequently, the higher chla observed in the ASP relative to the PIP, as well as the weak correspondence between chla and NPP in ASP, may reflect some sediment-driven optical effects rather than enhanced phytoplankton biomass or productivity alone. While our results are consistent with known differences in iron supply and mixed-layer dynamics between the two polynyas, the potential contribution of sediment-related bias cannot be ruled out and should be acknowledged when interpreting spatial contrasts in satellite chla on the Antarctic shelf."

**Minor comments**
L 26: edit "… in both chla and …" to "… in neither chla or…"
Corrected.

L70-74: this needs some clarification. Especially the part about vertical intrusions in PIP. What is meant by vertical intrusions? Do you mean upwelled mCDW?

We apologize for the lack of clarification. By vertical intrusion we intended to distinguish the small-scale upwelling of mCDW onto the shelf and beneath the ice shelf that occur more in the PIP, which would be called 'intrusions', as opposed to larger scale upwelling that would occur more in the ASP.

**Updated text:** "The PIP and ASP differ in their exposure to CDW and in local circulation: the ASP is more strongly influenced by upwelled modified CDW (mCDW) and glacial meltwater inputs, whereas in the PIP, the deep mCDW retains more of its original offshore characteristics, with vertical exchange only significantly occurring beneath the ice shelves, leading to a more stratified and less directly ventilated surface layer (Assmann et al., 2013; Dutrieux et al., 2014)"

L120-122: The winter mixed layer depth may be a more relevant metric for nutrient entrainment from depth. Furthermore, Fig. 1b shows summer MLD, so it is unclear what is meant by "mean mixed layer depth"
We agree with the reviewer. We have replaced Fig. 1b with the climatological winter mixed-layer depth (averaged for all years that we define as April-Sept, after the growing season and just before the start of the next one). What we meant by 'mean mixed-layer depth' was the climatological summer map (i.e originally all October-March averages from 1998 to 2017). We also accordingly updated the text.

**Updated text:** "The climatological winter mixed-layer depth (MLD) in the ASP is deeper (Fig. 1b), indicating that it may better entrain deeper sources of nutrients into the upper waters for the following phytoplankton growing season".

L 370: both → either
Corrected.

L 457: sediment → sediment-sourced dFe concentration
Corrected.

L 663: please add a reference
Thank you for pointing it out. We have added one reference that demonstrated that high surface biomass triggered by more iron brought to the surface from the meltwater pump does not necessarily imply high depth-integrated productivity.

Twelves, A. G., Goldberg, D. N., Henley, S. F., Mazloff, M. R., & Jones, D. C. (2021). Self-shading and meltwater spreading control the transition from light to iron limitation in an Antarctic coastal polynya. Journal of Geophysical Research: Oceans, 126, e2020JC016636. https://doi.org/10.1029/2020JC016636

L 748: add "suspected" underlying hydrographic drivers
Added.

L 791-793: this statement goes too far beyond the analysis presented. The results do not suggest long-term changes in the phytoplankton community composition. I suggest to rephrase or omit.
On second thoughts, we agree with the reviewer and decided to remove the sentence completely. Thank you.

L 797: tends → tend
Corrected.

L 802-811: this paragraph presents new results and should therefore be moved to the results section.
We have updated the manuscript as follow:

We added a brief statement in the method section about the ASL:
"Variability in the sea-ice landscape can be influenced by the Amundsen Sea Low in West Antarctica (ASL; Hosking et al., 2013; Turner et al., 2016). We therefore finally looked at the impact of the ASL and its potential influence on sea-ice variability. Monthly ASL indices (latitude, longitude, central and sector pressure) derived from ERA5 reanalysis data were obtained from the ASL climate index page (Hosking et al., 2016)."

We moved the results part in the Results section:
"Finally, we found on average weak spatial negative relationships between SIC and ASL latitude, longitude, mean sector and actual central pressure in both polynyas during the growing season (Supplementary Fig. S7), and only slightly significant in the eastern PIP."

We finally kept the original text in the discussion section:
"The weak relationships between the ASL indices and SIC might be owing to the seasonal variation of the ASL, where its position largely varies during summer, and its impact in shaping coastal sea ice is also greater during winter and autumn in the Amundsen-Bellingshausen region (Hosking et al., 2013). The lack of strong significant relationships overall does not allow us to conclude that the ASL plays an important role in shaping the coastal polynyas landscape and influencing chla variability."

L 851: delete "potential." The results demonstrate a robust and significant relationship between ice shelf melting and surface chla, so "potential" is unnecessary.
Corrected. Thank you.